# THE ACCUMULATION OF SCORE ESTIMATION ERROR IN DIFFUSION MODELS

## ABSTRACT

Diffusion models are widely used for high-quality generation, but their performance is sensitive to the accuracy of the estimated score. Our main results are established first in the setting where the forward process is initialized from a Gaussian mixture, where we derive Wasserstein bounds by leveraging the structure of the score and its Hessian. We then extend the analysis to general data distributions, where we provide a more general but looser upper bound. Our analysis reveals how discretization steps directly shape the accumulation of score estimation error, thereby explaining previously observed empirical phenomena regarding the advantage of certain discretization schedules. In addition, we show that, in the Gaussian setting, SDE samplers accumulate less error than ODE samplers in the small step-size regime, which explains their superior empirical performance. The result holds for both variance-preserving (VP) and variance-exploding (VE) diffusions.

## 1    INTRODUCTION

Diffusion models, also known as score-based generative models (Song et al., 2021), have become a leading paradigm for generative modeling, achieving state-of-the-art results in image synthesis (Rombach et al., 2022; Ramesh et al., 2022; Huang et al., 2025) and video generation (Bar-Tal et al., 2024; Blattmann et al., 2023). Diffusion models consist of two coupled processes: a forward process, which gradually perturbs data by adding noise, and a reverse process, which reconstructs data from Gaussian noise. The reverse dynamics require the score function, i.e., the gradient of the log-density of the perturbed distribution, denoted by $\nabla \log p_t(x)$ where $p_t$ is the marginal law of the forward process at time $t$.

Since the true score is intractable, it is approximated by training neural networks (Salimans & Ho; Song & Ermon, 2019; Ho et al., 2020), and the learned score is then employed to simulate the reverse process through discretized SDE or ODE solvers (see Section 2 for details).

There are two main sources of error in the reverse process: the discretization error, arising from numerical approximation of the dynamics, and the score estimation error, arising from approximating the true score with a learned network. Extensive prior work has focused on analyzing discretization error, i.e., the error introduced by numerically discretizing the reverse process (De Bortoli, 2022; Chen et al., 2023; Benton et al., 2024; Li & Cai, 2024; Li et al., 2025). In these analyses, the learned score $s_\theta(x, t)$ is typically assumed to approximate the true score $\nabla \log p_t(x)$ with a uniform $L^2$ error bounded by $\epsilon_0^2$, and the subsequent analysis focuses on the discretization error of the sampling method under the assumption of access to the ground-truth score.

However, existing analyses of score estimation error are rather coarse: its effect on the final distribution is usually bounded by terms of order $T\epsilon_0^2$ (Chen et al., 2023; Benton et al., 2024), which obscure the role of step-size allocation and fail to capture which regions of the time horizon contribute most critically. Prior work has demonstrated that the choice of step sizes has a significant impact on the quality of generated samples (Karras et al., 2022; Lu et al., 2022a; Sabour et al., 2024). In particular, although the estimation error at each step may be small, it propagates across the entire reverse trajectory and can substantially degrade sample quality. This issue is especially pronounced in regions of low signal-to-noise ratio (SNR), where empirical evidence shows that score approximation errors are relatively large (Nichol & Dhariwal, 2021; Wu et al., 2024).

Understanding how such errors accumulate under different discretization schemes is therefore essential for explaining the sensitivity of diffusion models to noise schedules and for developing more robust samplers. Motivated by this gap, our work develops a non-asymptotic analysis of score error propagation, yielding theoretical insights that explain observed schedule sensitivity and clarify the roles of discretization strategies and sampling formulations.

Our main contributions are summarized as follows:

- We derive stepwise Wasserstein bounds that preciselycharacterize how score estimation errors accumulate along the reverse dynamics.
- We provide a theoretical explanation for the empirical advantage of data-end-refining schedules such as cosine and uniform log-SNR, showing that they reduce error growth more effectively than linear schedules.
- We further clarify why SDE samplers empirically outperform ODE samplers: in the Gaussian setting, we show that the amplification factors in SDE updates are uniformly smaller, causing SDE dynamics to accumulate strictly less score–estimation error under the same discretization.

## 2 PRELIMINARIES

In this section we provide background on diffusion models, including the forward and reverse processes, score estimation, and sampling methods.

**Forward Process** The forward process gradually perturbs a clean data point $x_0 \sim p_0$, where $p_0$ is a distribution on $\mathbb{R}^d$. Its evolution is described by the stochastic differential equation

$$dX_t = \beta(t)X_t \, dt + \alpha(t) \, dW_t, \tag{1}$$

where $(W_t)_{t \geq 0}$ is a standard Brownian motion in $\mathbb{R}^d$, and we denote by $p_t$ the law of $X_t$ for each $t \in [0, T]$.

**Reverse Process** The reverse process reconstructs data by inverting the forward dynamics. It is initialized from $Y_0 \sim q_0$, where $q_0 = p_T$ is the terminal law of the forward process, and evolves back to a distribution $q_T$ close to the data distribution $p_0$. The reverse-time SDE is given by Anderson (1982); Song et al. (2021):

$$dY_t = \left(\beta(t)Y_t - \alpha(t)^2 \nabla \log p_t(Y_t)\right)dt + \alpha(t)d\overline{W}_t, \tag{2}$$

where $(\overline{W}_t)_{t \geq 0}$ is a time-reversed Brownian motion, and $\nabla \log p_t(x)$ denotes the score function of $p_t$. By construction, the forward and reverse processes are coupled through their marginals:

$$X_t \sim p_t \qquad \text{and} \qquad Y_t \sim q_t \text{ with } q_t = p_{T-t}.$$

In particular, the forward terminal distribution $p_T$ serves as the initialization $q_0$ for the reverse dynamics, and the reverse terminal distribution $q_T$ recovers the data distribution $p_0$.

**Score Estimation** In practice, the true score $\nabla \log p_t(x)$ is inaccessible since the marginal distribution $p_t$ is unknown. To address this, one trains a time-dependent neural network $s_\theta(x,t)$ using *denoising score matching* (DSM) (Vincent, 2011; Song & Ermon, 2019). The DSM objective is

$$\min_\theta \, \mathbb{E}_{t \sim U(0,T)} \, \mathbb{E}_{x_0 \sim p_0} \, \mathbb{E}_{x_t \sim p_{t|0}} \left[ \left\| s_\theta(x_t, t) - \nabla \log p_{t|0}(x_t) \right\|_2^2 \right],$$

so that the learned network $s_\theta$ provides an approximation of the true score function and can be used in place of $\nabla \log p_t$ in the reverse dynamics.

**Sampling Methods** Once the score network is trained, new samples are generated by simulating the reverse-time dynamics. This requires discretizing the reverse SDE or ODE. The specific form of the reverse process depends on the choice of coefficients $(\beta(t), \alpha(t))$ in the forward SDE equation 1.

Two standard formulations are widely used: the **variance-preserving (VP)** diffusion, with $\beta(t) = -1$ and $\alpha(t) = \sqrt{2}$ (Chen et al., 2023), and the **variance-exploding (VE)** diffusion, with $\beta(t) = 0$

and $\alpha(t) = \sqrt{2}$ (Song et al., 2021). For these two cases, the forward marginals admit closed-form conditionals:

$$p(x_t \mid x_0) = \begin{cases} \mathcal{N}\big(e^{-t}x_0, \ (1 - e^{-2t})I_d\big), & \text{VP}, \\ \mathcal{N}\big(x_0, \ 2tI_d\big), & \text{VE}. \end{cases}$$

We next introduce the time discretization used for simulating the reverse dynamics. Let $\{h_j\}_{j=0}^{K-1}$ with $h_j > 0$ denote a partition of $[0, T]$ into $K$ steps, and define the forward grid

$$t_k = \sum_{j=0}^{k-1} h_j, \qquad T = \sum_{j=0}^{K-1} h_j.$$

The reverse-time grid is simply the forward grid read backwards:

$$\tau_k = t_{K-1-k}, \qquad h_k^{\leftarrow} = h_{K-1-k}.$$

We illustrate the scheme using the exponential integrator (EI) method (Zhang & Chen, 2023). For the VP-SDE, the reverse update is

$$y_{k+1} = e^{h_k^{\leftarrow}} y_k + 2\big(e^{h_k^{\leftarrow}} - 1\big)\nabla \log p_{\tau_k}(y_k) + \sqrt{e^{2h_k^{\leftarrow}} - 1}\, z_k, \tag{3}$$

with initialization $y_0 \sim \mathcal{N}(0, I_d)$ and Gaussian noise $z_k \sim \mathcal{N}(0, I_d)$.

For the VE-SDE, the corresponding reverse update is

$$y_{k+1} = y_k + 2h_k^{\leftarrow} \nabla \log p_{\tau_k}(y_k) + \sqrt{2h_k^{\leftarrow}}\, z_k, \tag{4}$$

with initialization $y_0 \sim \mathcal{N}(0, 2TI_d)$ and $z_k \sim \mathcal{N}(0, I_d)$.

## 3 MAIN RESULTS

**Assumption 1** (Score Approximation Error). *Let*

$$e(x, t) := s_\theta(x, t) - \nabla \log p_t(x)$$

*denote the score approximation error at time $t$. We assume the following two mild regularity conditions:*

*(1) For each $t \in [0, T]$, the map $x \mapsto e(x, t)$ is $L_t$–Lipschitz:*

$$\|e(x, t) - e(y, t)\| \le L_t \|x - y\|, \qquad \forall x, y \in \mathbb{R}^d.$$

*(2) For each $t \in [0, T]$, the second moment of the error is finite:*

$$\mathbb{E}_{x \sim p_t}\big[\|e(x, t)\|^2\big] \le \varepsilon_t^2.$$

These conditions are mild and are satisfied by a broad class of practical score networks. In particular, we verify in Lemma 3 (Appendix B) that the Lipschitz requirement in Assumption 1 holds automatically when $p_0$ is a Gaussian mixture. The slice-wise $L^2$ boundedness is a standard assumption in the theoretical analysis of diffusion models (Chen et al., 2023; Benton et al., 2024).

Under these conditions, Assumption 1 guarantees that the deviation $e(y_k, \tau_k)$ at each reverse step is controlled both in magnitude and in its dependence on the state, which is exactly what is needed for our error-propagation analysis.

With Assumption 1, the perturbed reverse updates for VP/VE take the form

$$y_{k+1} = e^{h_k^{\leftarrow}} y_k + 2\big(e^{h_k^{\leftarrow}} - 1\big)\Big(\nabla \log p_{\tau_k}(y_k) + e(y_k, \tau_k)\Big) + \sqrt{e^{2h_k^{\leftarrow}} - 1}\, z_k, \tag{5}$$

with initialization $y_0 \sim \mathcal{N}(0, I_d)$. The corresponding baseline trajectory $\{y_k^{(0)}\}$ is obtained by removing the error term $e(y_k, \tau_k)$. For the VE-SDE, the update is

$$y_{k+1} = y_k + 2h_k^{\leftarrow}\Big(\nabla \log p_{\tau_k}(y_k) + e(\tau_k, y_k)\Big) + \sqrt{2h_k^{\leftarrow}}\, z_k, \tag{6}$$

with initialization $y_0 \sim \mathcal{N}(0, 2TI_d)$ for horizon $T > 0$. Again, the baseline sequence $\{y_k^{(0)}\}$ is defined analogously by removing $e(y_k, \tau_k)$. We denote by $p(y_K) := \mathcal{L}(y_K)$ and $p(y_K^{(0)}) := \mathcal{L}(y_K^{(0)})$ the terminal laws of the perturbed and baseline updates, respectively.

In the following sections, we characterize the Wasserstein distance induced by score perturbations. When the forward process is initialized from a Gaussian law $p(x_0)$, Section 3.1 derives the exact distance between $p(y_K)$ and $p(y_K^{(0)})$. Section 3.2 generalizes this to the case where $p(x_0)$ is a Gaussian mixture, and Section 3.3 further extends the analysis to arbitrary initial distributions $p(x_0)$.

## 3.1 GAUSSIAN DISTRIBUTION

Before presenting the general results, we first highlight the mechanism in the simple case where the initial distribution for the forward process is Gaussian: $p_0 = \mathcal{N}(\mu_0, \Sigma_0)$. Under the forward VP/VE processes the distribution remains Gaussian, and the score admits the closed form

$$\nabla_x \log p_t(x) = -\Sigma_t^{-1}(x - \mu_t), \tag{7}$$

where $(\mu_t, \Sigma_t)$ correspond to either VP or VE dynamics, as given in equation 8.

$$\begin{aligned} \mu^{\text{VP}}(t) &= e^{-t}\mu_i(0), & \Sigma_i^{\text{VP}}(t) &= e^{-2t}\Sigma(0) + (1 - e^{-2t})I_d, \\ \mu^{\text{VE}}(t) &= \mu(0), & \Sigma_i^{\text{VE}}(t) &= \Sigma(0) + 2tI_d. \end{aligned} \tag{8}$$

For a given discretization, define the operators

$$G_i(H) := \Big( \prod_{j=i+1}^{K-1} \big( \alpha_j I_d + \beta_j (H_j + L_{\tau_j} I_d) \big) \Big) \beta_i, \tag{9}$$

with $(\alpha_j, \beta_j) = (e^{h_j^{\leftarrow}}, 2(e^{h_j^{\leftarrow}} - 1))$ for VP-SDE and $(\alpha_j, \beta_j) = (1, 2h_j^{\leftarrow})$ for VE-SDE.

Then we have the following Theorem 3.1, which provides an upper bound on the Wasserstein distance between the perturbed and baseline terminal laws. The proof is given in Appendix A.

**Theorem 3.1.** *Under Assumption 1 and the Gaussian score representation equation 7,*

$$W_2^2\big(p(y_K), p(y_K^{(0)})\big) \leq \sum_{i=0}^{K-1} \|G_i(H)\|_{\text{op}}^2 \, \varepsilon_{\tau_i}^2, \tag{10}$$

*with $G_i(H)$ defined in equation 9 and $H = -\Sigma_{\tau_i}^{-1}$.*

**Remark.** *The bound equation 10 shows that the terminal Wasserstein error is determined by two main components: the amplification factors $G_i(H)$ and the local error magnitudes $\varepsilon_{\tau_i}$. The operators $G_i(H)$ depend on the discretization schedule as well as the curvature matrices $H_j = -\Sigma_{\tau_j}^{-1}$ and the Lipschitz constants $L_{\tau_j}$, which together control how perturbations are amplified along the reverse dynamics. Thus, both the geometry of $p_t$ and the Lipschitz behavior of the learned score govern how local errors accumulate over time.*

To further illustrate Theorem 3.1, consider the isotropic Gaussian $p_0 = \mathcal{N}(\mu_0, \sigma_0^2 I_d)$. Motivated by empirical findings that score errors are relatively large near the data end (small $t$) (Nichol & Dhariwal, 2021; Wu et al., 2024), we assume that the error profile $\{\varepsilon_t\}_{t \in [0,T]}$ is non-increasing in $t$, i.e.,

$$\varepsilon_t \geq \varepsilon_s \qquad \text{for all } 0 \leq t \leq s \leq T.$$

Hence Theorem 3.1 specializes, in the small step-size regime ($h_j^{\leftarrow} \ll 1$), to the approximation

$$W_2^2\big(p(y_K), p(y_K^{(0)})\big) \leq \sum_{0 \leq i \leq K-1} \left\| \beta_i \exp\Big( \sum_{j=i+1}^{K-1} h_j^{\leftarrow} \phi_{\tau_j} \Big) \right\|^2 \epsilon_{\tau_i}^2 \tag{11}$$

where

$$\phi_\tau = \begin{cases} 1 - 2c_\tau^{\text{VP}} + 2L_\tau, & \text{VP}, \\ -2c_\tau^{\text{VE}} + 2L_\tau, & \text{VE}, \end{cases} \qquad c_\tau^{\text{VP}} = \big(1 - (1 - \sigma_0^2)e^{-2\tau}\big)^{-1}, \quad c_\tau^{\text{VE}} = (\sigma_0^2 + 2\tau)^{-1}.$$

From equation 11, the error growth is governed by the amplification factors $\exp(\sum_{j=i+1}^{K-1} h_j^{\leftarrow} \phi_{\tau_j})$, which in turn depend on the coefficients $c_{\tau_i}$. Both VP and VE follow the same qualitative principle: refining the discretization (taking smaller $h_i^{\leftarrow}$) in regions where $\phi_{\tau_i}$ is large reduces amplification and decreases the accumulation of score errors. Importantly, in the noise end ($t \simeq T$), one typically has $\phi_{\tau_j} < 0$, so the factors $\exp(\sum h_j^{\leftarrow} \phi_{\tau_j})$ contribute a natural contraction that damps the error, and large steps can be taken without significant loss. By contrast, near the data end ($t \simeq 0$), $\epsilon_{\tau_i}$ is large, so bias terms may dominate. In this regime, smaller step sizes are required to mitigate error accumulation and prevent the bias from increasing the error.

Consequently, the overall implication is that one should use larger step sizes toward the noise end (near $t = T$), where errors are naturally damped, and smaller step sizes near the data end (near $t = 0$), where the bias is large. This conclusion is consistent with empirical findings on schedule design (Nichol & Dhariwal, 2021; Karras et al., 2022; Hang et al., 2024).

## 3.2 GAUSSIAN MIXTURES

The Gaussian case in Theorem 3.1 provides a clean closed-form expression where the linear structure of the score equation 7 leads directly to an exact Wasserstein error formula. This toy example illustrates the central mechanism by which score perturbations propagate through the dynamics.

We now extend the analysis to the more general and practically relevant case where the initial distribution is a mixture of Gaussians. In this setting, the score is no longer linear in $x$, yet the mixture structure still enables meaningful control of the induced Wasserstein error. Specifically, let

$$p_0(x) = \sum_{i=1}^{M} \pi_i \mathcal{N}(x; \mu_i(0), \Sigma_i(0)), \qquad \pi_i > 0, \ \sum_{i=1}^{M} \pi_i = 1, \ \Sigma_i(0) \succ 0, \tag{12}$$

and denote by $p_t$ the forward law at time $t$. As in the Gaussian case, under VP/VE diffusions each mixture component evolves according to equation 8, i.e.

$$p_t(x) = \sum_{i=1}^{M} \pi_i \mathcal{N}(x; \mu_i(t), \Sigma_i(t)).$$

To control the error propagation, we first introduce the exact pathwise Hessian average at step $k$:

$$H_k = \int_0^1 \nabla^2 \log p_{\tau_k}(y_k^{(0)} + t(y_k - y_k^{(0)})) \, dt. \tag{13}$$

Here $\nabla^2 \log p_{\tau_k}(x)$ is the Hessian of the log-density at time $\tau_k$ (equivalently, the Jacobian of the score $\nabla_x \log p_{\tau_k}(x)$). For a Gaussian mixture $p_{\tau_k}(x) = \sum_{m=1}^{M} \pi_m \mathcal{N}(x; \mu_m(\tau_k), \Sigma_m(\tau_k))$, it admits the decomposition

$$\nabla^2 \log p_{\tau_k}(x) = -\sum_{m=1}^{M} \gamma_m(x; \tau_k) \Sigma_m(\tau_k)^{-1} + \text{Cov}_{m \sim \gamma(\cdot \,|\, x; \tau_k)}[v_m(x; \tau_k)], \tag{14}$$

where

$$\gamma_m(x; \tau_k) := \frac{\pi_m \mathcal{N}(x; \mu_m(\tau_k), \Sigma_m(\tau_k))}{\sum_{j=1}^{M} \pi_j \mathcal{N}(x; \mu_j(\tau_k), \Sigma_j(\tau_k))}, \qquad v_m(x; \tau_k) := \Sigma_m(\tau_k)^{-1}(\mu_m(\tau_k) - x).$$

With $H_k$ defined in equation 13, we obtain an expression for the Wasserstein distance in the Gaussian-mixture setting analogous to Theorem 3.1:

$$W_2^2(p(y_K), p(y_K^{(0)})) \leq \sum_{i=0}^{K-1} \|G_i(H)\|_{op}^2 \, \epsilon_{\tau_i}^2 \tag{15}$$

where $G_i(\cdot)$ is defined in equation 9 and, by equation 13, $H = \{H_k\}_{k=0}^{K-1}$ denotes the stepwise Hessian averages along the coupled paths.

In practice, however, the exact pathwise Hessians $H_k$ in equation 13 are not available. We therefore introduce two computable surrogates: an *average surrogate*, obtained by weighting component

Hessians by their mixture weights, and a *dominant-component surrogate*, obtained by taking the Hessian of the most likely component at $y_k$:

$$\bar{H}_k^{\text{ave}} := -\sum_{m=1}^{M} \pi_m \Sigma_m(\tau_k)^{-1}, \quad \bar{H}_k^{\text{dom}} := -\Sigma_{i^\star(y_k)}(\tau_k)^{-1}, \quad i^\star(y_k) = \arg\max_{m \in [M]} \gamma_m(y_k; \tau_k).$$

(16)

We next show that replacing the exact $H_k$ with either surrogate still yields a valid Wasserstein error bound, with the guarantee depending on the tighter of the two choices.

**Theorem 3.2** (GM bound with surrogate Hessians). *Let $G_i(\cdot)$ be defined in equation 9. Under Assumption 1, and assuming the forward initial law is the Gaussian mixture in equation 12, the terminal laws of the perturbed and baseline updates satisfy*

$$W_2^2(p_K, p_K^{(0)}) \leq \min_{r \in \{\text{ave, dom}\}} \sum_{i=0}^{K-1} \left\| G_i(\bar{H}^{(r)}) \right\|_{op}^2 \epsilon_{\tau_i}^2 + \widehat{\Delta},$$

(17)

$$\widehat{\Delta} \leq C \left( \sum_{i:\tau_i \in I} \beta_i \sum_{j=i+1}^{K-1} \beta_j (d+2) \Lambda_j \right) \mathcal{S}_0,$$

*where*

$$\mathcal{S}_0 := \max_{0 \leq i \leq K-1} \epsilon_{\tau_i}, \qquad \Lambda_j := \max_{m \in [K]} \|\Sigma_m(\tau_j)^{-1}\|_{\text{op}}.$$

*Here $(\alpha_j, \beta_j)$ in $G_i(\cdot)$ are those of the chosen VP/VE sampler (cf. equation 9), and $C > 0$ is an absolute constant independent of $K$ and $d$.*

The proof is deferred to Appendix A.

**Remark.** *With Theorem 3.4 we obtain a similar implication as in Section 3.1. Near the data end ($t = 0$), the bias terms $\|\mu_{\tau_i}\|$ are large, whereas near the noise end ($t = T$) both surrogates $\bar{H}^{\text{ave}}$ and $\bar{H}^{\text{dom}}$ provide close approximations to the true score Hessian, as already discussed in the Gaussian setting. This indicates that small step sizes near the data end are crucial for controlling error accumulation, while larger step sizes can be safely adopted toward the noise end, thereby reducing the overall error in equation 17.*

To illustrate our theoretical results, we compare several step-size schedules with the Wasserstein error bounds predicted by Theorem 3.2 and the empirical performance obtained from DDPM sampling (Ho et al., 2020). The data distribution is a symmetric Gaussian mixture in $\mathbb{R}^{10}$,

$$p_0(x) = \tfrac{1}{2}\mathcal{N}(-1, I) + \tfrac{1}{2}\mathcal{N}(1, I).$$

In the experiments, the score function is learned by a neural network trained using denoising score matching. As shown in Figure 1, schedules that allocate *smaller* steps near the data end ($t \simeq 0$) achieve a *smaller* final $W_2$, consistent with our theoretical finding that error amplification is most sensitive in this region. Moreover, Theorem 3.2 produces bounds that preserve the same ordering across discretization schedules, providing an accurate theoretical characterization of the practical behavior observed during sampling. We also provide the experimental results with synthetic scores in Appendix D.

Beyond this synthetic setting, prior work on large-scale diffusion models (e.g., ImageNet $64 \times 64$ and CIFAR-10) has also reported that cosine-type schedules outperform linear schedules under the same pretrained score model (Nichol & Dhariwal, 2021). This empirical pattern is consistent with the preference suggested by our analysis.

We now turn to the choice of the surrogate Hessian $\bar{H}_k$. The appearance of the minimum in equation 17 reflects that, depending on the geometry of the Gaussian mixture, either the mixture-weighted surrogate $\bar{H}^{\text{ave}}$ or the dominant-component surrogate $\bar{H}^{\text{dom}}$ may yield a tighter control of the error.

We distinguish two regimes that guide the choice of the surrogate $\bar{H}_k$:

**Definition 1** (Small separation). *Define the mean separation*

$$\delta_\mu(t) := \max_{m \neq n} \left\| \Sigma_m(t)^{-1/2} (\mu_m(t) - \mu_n(t)) \right\|,$$

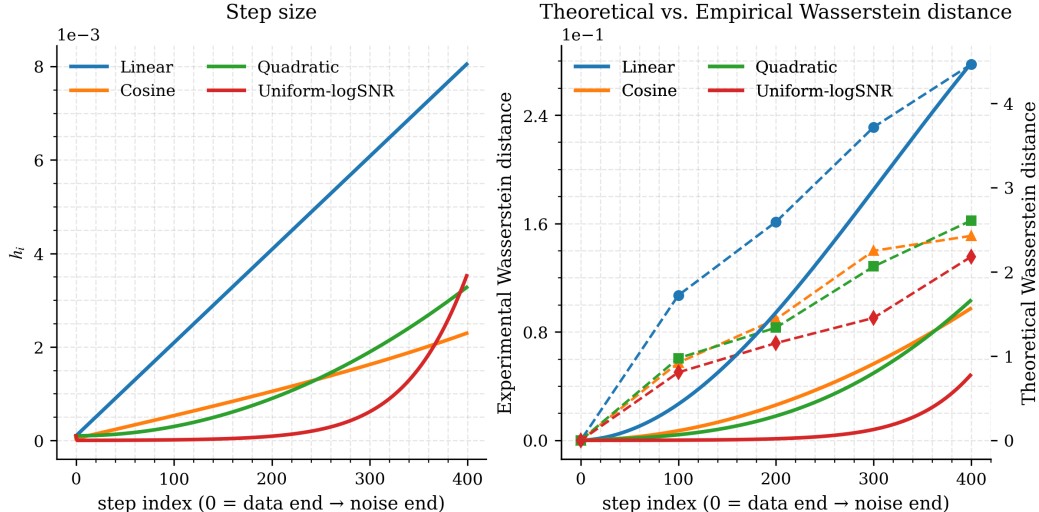

Figure 1: **Left:** Step-size profiles for several commonly used schedules (linear, quadratic, cosine, and uniform log–SNR). **Right:** Theoretical $W_2$ error predicted by Theorem 3.2 (solid lines) together with empirical estimates (dashed lines). Schedules that place *smaller* steps near the data end ($t \simeq 0$) yield a *smaller* final $W_2$, confirming that error amplification is most sensitive in this region. Moreover, the theoretical bounds closely track the empirical errors, capturing both their magnitudes and trends, which demonstrates the effectiveness of our analysis.

*and the covariance separation*

$$\delta_\Sigma(t) := \max_{m,n} \left\| \Sigma_m(t)^{1/2} \Sigma_n(t)^{-1} \Sigma_m(t)^{1/2} - I_d \right\|_{\mathrm{op}}.$$

*Let $\delta(t) := \max\{\delta_\mu(t), \delta_\Sigma(t)\}$. We say the mixture is in the* small separation regime *at time $t$ if $\delta(t) \ll 1$.*

**Definition 2** (Large separation). *For $x \in \mathbb{R}^d$, define the logits*

$$\ell_i(x) = \log \pi_i - \tfrac{1}{2} \log \det(2\pi \Sigma_i(t)) - \tfrac{1}{2}(x - \mu_i(t))^\top \Sigma_i(t)^{-1}(x - \mu_i(t)).$$

*Let $i^*(x) = \arg\max_i \ell_i(x)$ and the logit margin*

$$\kappa_t(x) := \min_{j \neq i^*(x)} \left( \ell_{i^*}(x) - \ell_j(x) \right).$$

*We say the mixture is in the* large separation regime *along a path $\{x_t\}$ if $\kappa_{\tau_k}(x_t) \geq \underline{\kappa} \gg 1$ for all $t \in [0, 1]$.*

We give Theorem 3.3 to refine Theorem 3.2 by adapting the surrogate Hessian according to the separation regime of the mixture. The proof of Theorem 3.3 is deffered to Appendix A.

**Theorem 3.3.** *Under the same setting as Theorem 3.2, let $p_{\tau_k}$ denote the marginal distribution of the forward process at time $\tau_k$. Define*

$$K_S := \max\{\, k : p_{\tau_k} \text{ lies in the small separation regime} \,\},$$

$$K_L := \min\{\, k : p_{\tau_k} \text{ lies in the large separation regime} \,\}.$$

*Then, with regime-adapted choices of $\bar{H}_k$, the error term $\widehat{\Delta}$ in equation 17 satisfies*

$$\widehat{\Delta} \leq \sum_{k=0}^{K_S} O\big(\delta(\tau_k)\big) \; + \; \sum_{k=K_L}^{K} O\big(e^{-\underline{\kappa}}\big)$$

$$+ \sum_{k=K_S+1}^{K_L-1} \left( \beta_k \sum_{j=k+1}^{K-1} \beta_j \, (d+2)\Lambda_j \right) \mathcal{S}_0.$$

(18)

This result highlights that the surrogate choice for $\bar{H}_k$ can be made adaptively: when the mixture is in the small separation regime, averaging across mixture components provides a reliable surrogate; when it is in the large separation regime, the dominant-component surrogate more closely matches the true Hessian. Both cases yield substantially sharper error control than the crude uniform bound. In particular, the error contributions scale as $O(\delta(\tau_i))$ in small-separation regions and decay exponentially in $\underline{\kappa}$ in large-separation regions. Only in intermediate cases where the mixture is neither clearly separated nor overlapping, do we still have the coarse $(d+2)\Lambda_j$ bound.

Moreover, this refinement connects directly to the properties of the initial distribution $p_0(x)$. If $p_0(x)$ is in the small separation regime, then $\widehat{\Delta}$ can be controlled at order $O(\delta)$. If $p_0(x)$ is instead in the large separation regime, and the error perturbations $e_\tau$ are concentrated only near the data end (i.e., at small diffusion times), then $\widehat{\Delta}$ can be controlled at order $O(e^{-\underline{\kappa}})$. Consequently, in these settings the leading terms in equation 17 provide an accurate reflection of the Wasserstein discrepancy, with $\widehat{\Delta}$ reduced to a negligible correction.

### 3.3 GENERAL DISTRIBUTIONS

The Gaussian and Gaussian-mixture cases show that structural assumptions on the data distribution can yield sharp and interpretable error bounds. For completeness, we now state a more general result that applies to arbitrary data distributions without requiring such assumptions.

**Theorem 3.4.** *Consider the VP/VE reverse recursions equation 5–equation 6 under synchronous coupling. Let $p_K = \mathcal{L}(y_K)$ and $p_K^{(0)} = \mathcal{L}(y_K^{(0)})$ denote the terminal laws of the perturbed and baseline updates, respectively. Under Assumption 1, the terminal Wasserstein deviation satisfies*

$$W_2^2(p_K, p_K^{(0)}) \leq \sum_{i:\,\tau_i \in I} \|G_i(H)\|_{\mathrm{op}}^2 \varepsilon_{\tau_i}^2, \tag{19}$$

*where $G_i(H)$ is defined in equation 9, with $H$ taken as*

$$H_j = \begin{cases} \dfrac{d}{\sigma_{\mathrm{VP}}(\tau_j)^2}\, I_d, & \text{VP-SDE,} \\[2ex] \dfrac{d}{\sigma_{\mathrm{VE}}(\tau_j)^2}\, I_d, & \text{VE-SDE,} \end{cases}$$

*and $\sigma_{\mathrm{VP}}(\tau) = \sqrt{1 - e^{-2\tau}}$, $\sigma_{\mathrm{VE}}(\tau) = \sqrt{2\tau}$ denote the forward smoothing scales.*

Theorem 3.4 shows that even without structural assumptions, a non-asymptotic Wasserstein bound can be obtained by controlling the curvature of the forward marginals through their smoothing scales. This bound is necessarily conservative: as $\tau \to 0$, the forward variance vanishes and $\sigma_{\mathrm{VP/VE}}(\tau) \to 0$, causing $H$ to blow up. Near the noise end, $\phi_T^{\mathrm{VP}} \approx -1$ and $\phi_T^{\mathrm{VE}} = -1/T < 0$, so amplification is weak and large steps are safe. Near the data end, curvature can be large, and small steps are essential. For these reasons, Theorem 3.4 is stated without structural assumptions: it serves as a worst-case baseline showing that the data-end region is inherently more sensitive to score-estimation errors.

When the data distribution has a Lipschitz score, the curvature terms $H_j$ in equation 19 can be further tightened. In particular, $H_j$ admits a uniform bound for all sufficiently small times, leading to a sharper bound in this regime. See Corollary B.1 in Appendix A for details.

### 3.4 EXTENSION TO PF-ODE

We now extend the result to the probability-flow ODE (PF-ODE) formulation of diffusion models. Equivalently, the reverse dynamics for equation 2 can be written as a probability-flow ODE with the same marginals Song et al. (2021):

$$dY_t^\leftarrow = \left(\beta(t)Y_t^\leftarrow - \tfrac{1}{2}\alpha(t)^2 \nabla_{\boldsymbol{x}} \log p_t(Y_t^\leftarrow)\right) dt. \tag{20}$$

For concreteness, consider the VP case, whose reverse update reads

$$y_{k+1} = e^{h_k^\leftarrow} y_k + \left(e^{h_k^\leftarrow} - 1\right)\left(s_{\tau_k}(y_k) + e(y_k, \tau_k)\right),$$

with $y_0 \sim \mathcal{N}(0, I_d)$. The baseline trajectory $\{y_k^{(0)}\}$ is obtained by removing $e(y_k, \tau_k)$.

**Corollary 3.1.** *Under the same setting and notation as Theorem 3.2 (in particular $G_i(H)$ as in equation 9), the probability-flow ODE discretization satisfies*

$$W_2^2\big(p_K, p_K^{(0)}\big) \leq \sum_{i=0}^{K-1} \|G_i(H)\|_{op}^2 \, \epsilon_{\tau_i}^2,$$

(21)

*where the only change relative to the SDE case lies in the amplification coefficients in $G_i(\cdot)$:*

$$(\alpha_j, \beta_j) = \begin{cases} \big(e^{h_j^{\leftarrow}}, \, e^{h_j^{\leftarrow}} - 1\big), & \text{VP-PF-ODE}, \\ \big(1, \, h_j^{\leftarrow}\big), & \text{VE-PF-ODE}. \end{cases}$$

**Remark.** *This extension shows that our framework applies uniformly to both SDE- and ODE-based samplers. The bias–variance decomposition of the Wasserstein error remains unchanged, and the only difference arises from the amplification coefficients $(\alpha_j, \beta_j)$ encoded in $G_i(H)$.*

**SDE vs. ODE.** The resulting amplification factors are

$$\alpha_j - \beta_j c_{\tau_j} \approx \begin{cases} 1 + h_j^{\leftarrow}(1 - 2c_{\tau_j}), & \text{SDE}, \\ 1 + h_j^{\leftarrow}(1 - c_{\tau_j}), & \text{ODE}. \end{cases}$$

In the Gaussian setting, the curvature coefficient satisfies $c_{\tau_j} > 0$ for all $\tau_j$, so that $1 - 2c_{\tau_j} < 1 - c_{\tau_j}$. Consequently, the linearized amplification factor

$$\phi_{\tau_j} = \alpha_j - \beta_j c_{\tau_j}$$

is uniformly smaller for the SDE update than for the corresponding ODE update. This implies that each SDE step attenuates score-estimation error more strongly than its ODE counterpart. In the regime of sufficiently small step sizes, the resulting reverse recursion therefore exhibits strictly weaker cumulative amplification of score errors. This provides a principled explanation—within the Gaussian framework—for the empirically observed superiority of SDE-based samplers over ODE-based samplers in terms of sample quality (Lu et al., 2022b; Guo et al., 2023; Nie et al., 2024).

## 4 CONCLUSION

In this work, we analyzed how score estimation errors propagate through the reverse dynamics of diffusion models for both VP and VE processes under reverse SDE and PF-ODE. Starting from the Gaussian case, Theorem 3.1 provided an upper bound on the Wasserstein distance induced by score error, highlighting how discretization steps and the covariance jointly govern error accumulation. For Gaussian mixtures, Theorem 3.2 established a general bound, which can be further tightened under small- or large-separation conditions, thereby adapting to the geometry of the mixture components. Finally, Theorem 3.4 extended the framework to arbitrary data distributions, offering distribution-free but necessarily conservative guarantees. We also give refined bounds under smoothness assumptions on the data distribution in Corollary B.1.

Our analysis provides concrete insights into step-size allocation. Near the data end ($t = 0$), where bias is most pronounced, finer discretization is essential to suppress error accumulation, whereas near the noise end ($t = T$) larger steps can be safely used since amplification is weaker. This explains the empirical success of cosine and uniform log-SNR schedules compared to linear ones (Nichol & Dhariwal, 2021; Karras et al., 2022; Hang et al., 2024). Moreover, our results clarify why, in the Gaussian setting, SDE-based samplers accumulate less error than ODE-based samplers in the fine discretization regime, thereby providing a theoretical explanation for their empirical advantage

**Future Work.** This work has focused on how discretization schedules influence the propagation of score-estimation errors during sampling. An important next step is to extend this perspective to the training stage, where the choice of noise schedule also plays a critical role in learning the score function (Hang et al., 2023; Lin et al., 2024). Developing a unified end-to-end analysis that simultaneously accounts for both training and sampling schedules could provide a deeper theoretical foundation, especially since the theoretical impact of discretization schedules on training error remains largely unexplored. In particular, connecting our sampling-side accumulation bound with training-side guarantees would require time-resolved estimates of the score-estimation error at each noise level, which remains an open challenge.

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

## LLM USAGE STATEMENT

We used large language models (LLMs) only for grammar checking, typo correction, and polishing the writing. They were not used for other part of this work.

## A  PROOF OF MAIN RESULTS

*Proof of Theorem 3.1.* Let $\Delta_k := y_k - y_k^{(0)}$ under synchronous coupling, so the Gaussian noises of the perturbed and baseline recursions are identical. Subtracting the baseline update from the perturbed update and using the Gaussian score representation $\nabla \log p_{\tau_k}(x) = Hx + b_{\tau_k}$ with $H = -\Sigma_{\tau_k}^{-1}$ yields

$$\Delta_{k+1} = (\alpha_k I_d + \beta_k H)\Delta_k + \beta_k\big(e(\tau_k, y_k) - e(\tau_k, y_k^{(0)})\big) + \beta_k e(\tau_k, y_k^{(0)}).$$

By Assumption 1(1),

$$\|e(\tau_k, y_k) - e(\tau_k, y_k^{(0)})\| \leq L_{\tau_k}\|\Delta_k\|.$$

Thus,

$$\Delta_{k+1} = \widetilde{M}_k \Delta_k + \beta_k e(\tau_k, y_k^{(0)}), \qquad \widetilde{M}_k := \alpha_k I_d + \beta_k(H + L_{\tau_k} I_d).$$

Iterating from $\Delta_0 = 0$ gives the explicit expansion

$$\Delta_K = \sum_{i=0}^{K-1} G_i(H)\, e(\tau_i, y_i^{(0)}),$$

where $G_i(H)$ is exactly the matrix product defined in equation 9.

Using Cauchy–Schwarz and Assumption 1(2),

$$\mathbb{E}\|\Delta_K\|^2 = \mathbb{E}\Big\|\sum_{i:\,\tau_i \in I} G_i(H)\, e(\tau_i, y_i^{(0)})\Big\|^2$$

$$\leq \sum_{i:\,\tau_i \in I} \|G_i(H)\|_{\mathrm{op}}^2\, \mathbb{E}\|e(\tau_i, y_i^{(0)})\|^2$$

$$\leq \sum_{i:\,\tau_i \in I} \|G_i(H)\|_{\mathrm{op}}^2\, \varepsilon_{\tau_i}^2.$$

**Wasserstein distance.**  Under synchronous coupling,

$$W_2^2\big(p(y_K), p(y_K^{(0)})\big) \leq \mathbb{E}\|\Delta_K\|^2,$$

which together with the bound above establishes equation 10.  □

*proof of Theorem 3.2.* Let $\Delta_K := y_K - y_K^{(0)}$ under the synchronous coupling. The perturbation recursion unrolls as

$$\Delta_K = \sum_{i=0}^{K-1} G_i(H) e_{\tau_i},$$

where we abbreviate $e_{\tau_i} := e(\tau_i, y_i^{(0)})$ for simplicity. Add and subtract the surrogate gains $G_i(\bar{H})$:

$$\Delta_K = \underbrace{\sum_{i=0}^{K-1} G_i(\bar{H})e_{\tau_i}}_{=:S_1} + \underbrace{\sum_{i=0}^{K-1} \big(G_i(H) - G_i(\bar{H})\big)e_{\tau_i}}_{=:S_2}.$$

Hence

$$W_2^2\big(p_K, p_K^{(0)}\big) = \mathbb{E}\|\Delta_K\|^2 \leq 2\mathbb{E}\|S_1\|^2 + 2\mathbb{E}\|S_2\|^2.$$

*Control of $S_1$.* Independence across $i$ and $\mathbb{E}[e_{\tau_i}] = 0$ give

$$\mathbb{E}\|S_1\|^2 = \sum_{i=0}^{K-1} \left\|G_i(\bar{H})\right\|_{op}^2 \epsilon_{\tau_i}^2$$

*Control of $S_2$.* By expanding the product in equation 9 and using submultiplicativity,

$$G_i(H) - G_i(\bar{H}) = \left( \prod_{\ell=i+1}^{K-1} \left( \alpha_\ell I_d + \beta_\ell(H_\ell + L_\ell I_d) \right) - \prod_{\ell=i+1}^{K-1} \left( \alpha_\ell I_d + \beta_\ell(\bar{H}_\ell + L_\ell I_d) \right) \right) \beta_i$$

$$= \sum_{j=i+1}^{K-1} \left( \prod_{\ell=j+1}^{K-1} \left( \alpha_\ell I_d + \beta_\ell(H_\ell + L_\ell I_d) \right) \right) \beta_j (H_j - \bar{H}_j) \left( \prod_{\ell=i+1}^{j-1} \left( \alpha_\ell I_d + \beta_\ell(\bar{H}_\ell + L_\ell I_d) \right) \right) \beta_i.$$

*Taking operator norms and using submultiplicativity:*

$$\|G_i(H) - G_i(\bar{H})\|_{op} \le \sum_{j=i+1}^{K-1} \left( \prod_{\ell=j+1}^{K-1} \|\alpha_\ell I_d + \beta_\ell(H_\ell + L_\ell I_d)\|_{op} \right) \beta_j \|H_j - \bar{H}_j\|_{op}$$

$$\times \left( \prod_{\ell=i+1}^{j-1} \|\alpha_\ell I_d + \beta_\ell(\bar{H}_\ell + L_\ell I_d)\|_{op} \right) \beta_i.$$

Assume there exists a constant $C_0 \ge 1$ such that for all relevant $\ell$, $\|\alpha_\ell I_d + \beta_\ell(H_\ell + L_\ell I_d)\|_{op} \le C_0$ and $\|\alpha_\ell I_d + \beta_\ell(\bar{H}_\ell + L_\ell I_d)\|_{op} \le C_0$. Then each product is bounded by a constant that we absorb into $C$, yielding

$$\|G_i(H) - G_i(\bar{H})\|_{op} \le C \beta_i \sum_{j=i+1}^{K-1} \beta_j \|H_j - \bar{H}_j\|_{op}.$$

Define

$$\mathcal{S}_0 := \max_{0 \le i \le K-1} \epsilon_{\tau_i}$$

Hence

$$\mathbb{E}\|S_2\| = \mathbb{E}\left\| \sum_{i=0}^{K-1} \left( G_i(H) - G_i(\bar{H}) \right) e_{\tau_i} \right\|$$

$$\le \sum_{i=0}^{K-1} \mathbb{E}\|G_i(H) - G_i(\bar{H})\|_{op} \, \mathbb{E}\|e_{\tau_i}\|$$

$$\le C \sum_{i=0}^{K-1} \left( \beta_i \sum_{j=i+1}^{K-1} \beta_j \mathbb{E}\|H_j - \bar{H}_j\|_{op} \right) \mathbb{E}\|e_{\tau_i}\|$$

$$\le C \left( \sum_{i=0}^{K-1} \beta_i \sum_{j=i+1}^{K-1} \beta_j (d+2) \Lambda_j \right) \mathcal{S}_0$$

The last inequality comes from Lemma 5: $\mathbb{E}\|H_j - \bar{H}_j\|_{op} \le (d+2)\Lambda_j$, we complete the proof. $\square$

*Proof of Theorem 3.3.* The argument follows the same structure as the proof of Theorem 3.4. In addition, by Lemma 6 and Lemma 7, we can control the deviation $\|H_i - \bar{H}_i\|$ depending on the regime of $p_{\tau_i}$: in the *small separation* regime the deviation is $O(\delta(\tau_i))$, while in the *large separation* regime it is $O(e^{-\frac{\kappa}{\tau}})$. Combining these bounds with the general estimate in Theorem 3.4 yields inequality equation 18. $\square$

*Proof of Theorem 3.4.* Let $\Delta_k := y_k - y_k^{(0)}$ under synchronous coupling. Subtracting the baseline update from the perturbed update gives

$$\Delta_{k+1} = a_k \Delta_k + b_k \left( \nabla \log p_{\tau_k}(y_k) - \nabla \log p_{\tau_k}(y_k^{(0)}) \right) + b_k \, e(\tau_k, y_k),$$

with $(a_k, b_k) = (e^{h_k^{\leftarrow}}, 2(e^{h_k^{\leftarrow}} - 1))$ for VP and $(a_k, b_k) = (1, 2h_k^{\leftarrow})$ for VE. By the mean–value representation,

$$\nabla \log p_{\tau_k}(y_k) - \nabla \log p_{\tau_k}(y_k^{(0)}) = H_k \Delta_k, \quad H_k := \int_0^1 \nabla^2 \log p_{\tau_k}(y_k^{(0)} + t\Delta_k) \, dt.$$

Lemma 1 implies

$$\mathbb{E}\|H_k\|_{\mathrm{op}} \leq \frac{d+1}{\sigma^2(\tau_k)} =: C_k.$$

Decomposing $e(\tau_k, y_k)$ as

$$e(\tau_k, y_k) = e(\tau_k, y_k^{(0)}) + \big(e(\tau_k, y_k) - e(\tau_k, y_k^{(0)})\big)$$

and using Assumption 1(1) yields

$$\|e(\tau_k, y_k) - e(\tau_k, y_k^{(0)})\| \leq L_{\tau_k}\|\Delta_k\|.$$

Hence

$$\Delta_{k+1} = \widetilde{M}_k \Delta_k + b_k e(\tau_k, y_k^{(0)}), \qquad \widetilde{M}_k := a_k I_d + b_k(H_k + L_{\tau_k} I_d),$$

and

$$\mathbb{E}\|\widetilde{M}_k\|_{\mathrm{op}} \leq a_k + b_k(C_k + L_{\tau_k}) =: \alpha_k.$$

Iterating from $\Delta_0 = 0$,

$$\Delta_K = \sum_{i:\, \tau_i \in I} G_i(H)\, e(\tau_i, y_i^{(0)}), \qquad G_i(H) := \Big(\prod_{j=i+1}^{K-1} \widetilde{M}_j\Big) b_i.$$

By Cauchy–Schwarz and Assumption 1(2),

$$\mathbb{E}\|\Delta_K\|^2 \leq \sum_{i:\, \tau_i \in I} \|G_i(H)\|_{\mathrm{op}}^2 \, \mathbb{E}\|e(\tau_i, y_i^{(0)})\|^2 \leq \sum_{i:\, \tau_i \in I} \|G_i(H)\|_{\mathrm{op}}^2 \varepsilon_{\tau_i}^2.$$

Finally, synchronous coupling gives

$$W_2^2(p_K, p_K^{(0)}) \leq \mathbb{E}\|\Delta_K\|^2,$$

establishing equation 19. $\qquad\square$

## B  USEFUL LEMMAS

**Lemma 1** (Expected operator–norm Hessian). *Let $X = \mu X_0 + \sigma Z$ with $Z \sim \mathcal{N}(0, I_d)$ independent of an arbitrary $X_0$ in $\mathbb{R}^d$, and let $p_{\mu,\sigma}$ be the density of $X$. Then*

$$\mathbb{E}\left\|\nabla^2 \log p_{\mu,\sigma}(X)\right\|_{\mathrm{op}} \leq \frac{d+1}{\sigma^2}.$$

*Proof of Lemma 1.* For $X = \mu X_0 + \sigma Z$ with density $p_{\mu,\sigma}$, differentiating the Gaussian-smoothed density under the integral (justified by dominated convergence for the Gaussian kernel) yields, for every $x \in \mathbb{R}^d$,

$$\nabla \log p_{\mu,\sigma}(x) = \frac{1}{\sigma^2}\Big(\mu \mathbb{E}[X_0 \mid X{=}x] - x\Big), \tag{22}$$

$$\nabla^2 \log p_{\mu,\sigma}(x) = \frac{1}{\sigma^4}\mathrm{Cov}(\mu X_0 \mid X{=}x) - \frac{1}{\sigma^2}I_d. \tag{23}$$

From equation 23 and $\|A\|_{\mathrm{op}} \leq \mathrm{tr}\,(A)$ for $A \succeq 0$,

$$\left\|\nabla^2 \log p_{\mu,\sigma}(x)\right\|_{\mathrm{op}} \leq \frac{1}{\sigma^4}\mathrm{tr}\,(\mathrm{Cov}(\mu X_0 \mid X{=}x)) + \frac{1}{\sigma^2}.$$

Taking expectation over $X$ and using the Bayes-risk optimality of the conditional mean,

$$\mathbb{E} \operatorname{tr}\left(\operatorname{Cov}(\mu X_0 \mid X)\right) = \mathbb{E}\mathbb{E}\Big[\big\|\mu X_0 - \mathbb{E}[\mu X_0 \mid X]\big\|^2\Big|X\Big] \le \mathbb{E}\big\|\mu X_0 - X\big\|^2.$$

Since $X = \mu X_0 + \sigma Z$ with $Z \sim \mathcal{N}(0, I_d)$ independent of $X_0$, we have

$$\mathbb{E}\big\|\mu X_0 - X\big\|^2 = \mathbb{E}\big\| - \sigma Z\big\|^2 = \sigma^2\mathbb{E}\|Z\|^2 = d\sigma^2.$$

Therefore,

$$\mathbb{E}\big\|\nabla^2 \log p_{\mu,\sigma}(X)\big\|_{\mathrm{op}} \le \frac{1}{\sigma^4}d\sigma^2 + \frac{1}{\sigma^2} = \frac{d+1}{\sigma^2}.$$

$\square$

**Lemma 2** (Universal expectation bound for Gaussian mixtures)**.** *Let* $p_t(x) = \sum_{m=1}^{K} \pi_m \mathcal{N}\big(x; \mu_m(t), \Sigma_m(t)\big)$ *and define*

$$\gamma_m(x) = \frac{\pi_m \varphi_m(x)}{p_t(x)}, \qquad v_m(x) = \Sigma_m(t)^{-1}\big(\mu_m(t) - x\big).$$

*Then, for* $X \sim p_t$,

$$\mathbb{E}\big\|\nabla^2 \log p_t(X)\big\|_{\mathrm{op}} \le \sum_{m=1}^{K} \pi_m\big\|\Sigma_m(t)^{-1}\big\|_{\mathrm{op}} + \sum_{m=1}^{K} \pi_m \operatorname{tr}\big(\Sigma_m(t)^{-1}\big).$$

*In particular, since* $\operatorname{tr}(A) \le d\|A\|_{\mathrm{op}}$,

$$\mathbb{E}\big\|\nabla^2 \log p_t(X)\big\|_{\mathrm{op}} \le (d+1)\sum_{m=1}^{K} \pi_m\big\|\Sigma_m(t)^{-1}\big\|_{\mathrm{op}} \le (d+1)\max_m\big\|\Sigma_m(t)^{-1}\big\|_{\mathrm{op}}.$$

*Proof of Lemma 2.* From the mixture Hessian identity,

$$\nabla^2 \log p_t(x) = -\sum_m \gamma_m(x)\Sigma_m(t)^{-1} + \operatorname{Cov}_{m\sim\gamma(\cdot|x)}\big[v_m(x)\big],$$

hence for any $x$,

$$\big\|\nabla^2 \log p_t(x)\big\|_{\mathrm{op}} \le \Big\|\sum_m \gamma_m(x)\Sigma_m(t)^{-1}\Big\|_{\mathrm{op}} + \mathbb{E}_{\gamma(\cdot|x)}\big\|v_m(x)\big\|^2.$$

*First term.* By triangle inequality, $\|\sum_m \gamma_m(x)\Sigma_m(t)^{-1}\|_{\mathrm{op}} \le \sum_m \gamma_m(x)\|\Sigma_m(t)^{-1}\|_{\mathrm{op}}$. Taking $\mathbb{E}$ in $X \sim p_t$ and using $\mathbb{E}[\gamma_m(X)] = \pi_m$ gives

$$\mathbb{E}\Big\|\sum_m \gamma_m(X)\Sigma_m(t)^{-1}\Big\|_{\mathrm{op}} \le \sum_m \pi_m\|\Sigma_m(t)^{-1}\|_{\mathrm{op}}.$$

*Second term.* By the law of total expectation under the generative model $M \sim \{\pi_m\}$, $X|M = m \sim \mathcal{N}(\mu_m(t), \Sigma_m(t))$,

$$\mathbb{E}_X\mathbb{E}_{\gamma(\cdot|X)}\big\|v_m(X)\big\|^2 = \mathbb{E}_{M,X}\big\|\Sigma_M(t)^{-1}\big(\mu_M(t) - X\big)\big\|^2.$$

Condition on $M = m$: $\mu_m(t) - X \sim \mathcal{N}(0, \Sigma_m(t))$, so

$$\mathbb{E}\Big[\big\|\Sigma_m(t)^{-1}\big(\mu_m(t) - X\big)\big\|^2\Big|M = m\Big] = \operatorname{tr}\big(\Sigma_m(t)^{-1}\big).$$

Averaging over $m$ with weights $\pi_m$ yields $\mathbb{E}_X\mathbb{E}_{\gamma(\cdot|X)}\|v_m(X)\|^2 = \sum_m \pi_m \operatorname{tr}\big(\Sigma_m(t)^{-1}\big)$.

Combine the two bounds to obtain the stated inequality. The final display follows from $\operatorname{tr}(A) \le d\|A\|_{\mathrm{op}}$ and $\sum_m \pi_m a_m \le \max_m a_m$. $\square$

**Lemma 3** (Lipschitz score error under Gaussian-mixture marginals). *Assume*

$$p_0(x) = \sum_{k=1}^{K} \pi_k \, \mathcal{N}(x; m_k, \Sigma_k), \qquad \pi_k > 0, \ \sum_k \pi_k = 1, \ \Sigma_k \succ 0.$$

*Let $(p_t)_{t \in (0,T]}$ be the forward marginals of a VP/VE diffusion, so that for each $t > 0$,*

$$p_t(x) = \sum_{k=1}^{K} \pi_k \, \mathcal{N}\big(x; m_k(t), \Sigma_k(t)\big), \qquad \Sigma_k(t) \succ 0.$$

*Then for every $t \in (0, T]$:*

*(a) The score $\nabla \log p_t(x)$ is globally Lipschitz in $x$, i.e.,*

$$\|\nabla \log p_t(x) - \nabla \log p_t(y)\| \leq L_t^{\star} \, \|x - y\|, \qquad \forall x, y \in \mathbb{R}^d$$

*for some finite constant $L_t^{\star} < \infty$.*

*(b) If the learned score $s_\theta(\cdot, t)$ is $L_t^\theta$–Lipschitz in $x$, then the score error*

$$e(t, x) := s_\theta(x, t) - \nabla \log p_t(x)$$

*is $L_t$–Lipschitz with*

$$L_t \ \leq \ L_t^\theta + L_t^{\star}.$$

*Proof.* Since $p_0$ is a finite Gaussian mixture, we may write

$$p_0(x) = \sum_{k=1}^{K} \pi_k \, \mathcal{N}(x; \mu_k(0), \Sigma_k(0)).$$

Under the VP/VE forward dynamics, each component evolves into another Gaussian with mean and covariance given by Eq. equation 24:

$$p_t(x) = \sum_{k=1}^{K} \pi_k \, \mathcal{N}\big(x; \mu_k(t), \Sigma_k(t)\big), \qquad t > 0,$$

where

$$\begin{aligned}
\mu_k^{\mathrm{VP}}(t) &= e^{-t} \, \mu_k(0), & \Sigma_k^{\mathrm{VP}}(t) &= e^{-2t} \Sigma_k(0) + \big(1 - e^{-2t}\big) I_d, \\
\mu_k^{\mathrm{VE}}(t) &= \mu_k(0), & \Sigma_k^{\mathrm{VE}}(t) &= \Sigma_k(0) + 2t I_d.
\end{aligned} \tag{24}$$

For any fixed $t > 0$, all component covariances $\Sigma_k(t)$ are strictly positive definite with eigenvalues uniformly bounded below by a constant $c_t > 0$. Each component density $\varphi_k(x) = \mathcal{N}(x; \mu_k(t), \Sigma_k(t))$ is smooth and strongly log-concave, and its Hessian $\nabla^2 \log \varphi_k(x)$ is a bounded matrix whose operator norm depends only on $\Sigma_k(t)$.

Let

$$p_t(x) = \sum_{k=1}^{K} \pi_k \varphi_k(x), \qquad w_k(x) = \frac{\pi_k \varphi_k(x)}{p_t(x)},$$

so that

$$\nabla \log p_t(x) = \sum_{k=1}^{K} w_k(x) \, \nabla \log \varphi_k(x).$$

Differentiating,

$$\nabla^2 \log p_t(x) = \sum_{k=1}^{K} w_k(x) \, \nabla^2 \log \varphi_k(x) + \mathrm{Cov}_{w(x)}\big(\nabla \log \varphi_k(x)\big),$$

where both terms are bounded uniformly in $x$. Hence

$$\sup_{x \in \mathbb{R}^d} \|\nabla^2 \log p_t(x)\|_{\mathrm{op}} < \infty.$$

By the mean-value theorem,

$$\|\nabla \log p_t(x) - \nabla \log p_t(y)\| \le L_t^\star \|x - y\|,$$

for some finite constant $L_t^\star$ depending only on $t$ and the mixture parameters. This establishes that $\nabla \log p_t$ is globally Lipschitz.

Finally, suppose the learned score $s_\theta(\cdot, t)$ is $L_t^\theta$–Lipschitz in $x$, i.e.,

$$\|s_\theta(x, t) - s_\theta(y, t)\| \le L_t^\theta \|x - y\|, \qquad \forall\, x, y.$$

Define the score error $e(t, x) = s_\theta(x, t) - \nabla \log p_t(x)$. Then for any $x, y$,

$$\|e(t, x) - e(t, y)\| \le \|s_\theta(x, t) - s_\theta(y, t)\| + \|\nabla \log p_t(x) - \nabla \log p_t(y)\|.$$

Using the Lipschitz constant $L_t^\star$ established above for $\nabla \log p_t$, we obtain

$$\|e(t, x) - e(t, y)\| \le (L_t^\theta + L_t^\star)\, \|x - y\|.$$

Thus $e(t, \cdot)$ is $L_t$–Lipschitz with

$$L_t \;\le\; L_t^\theta + L_t^\star,$$

completing the proof. $\qquad\square$

To further sharpen the behavior of the general bound in Theorem 3.4 as $t \to 0$, we analyze the case where the data distribution $p_0$ has a Lipschitz score function. Under this additional smoothness, the next lemma (Lemma 4) shows that the operator norm of the Hessian of $\log p_t$ remains uniformly controlled by that of $\log p_0$, and in particular does not exhibit the $1/t$ blow-up present in the distribution-free bound.

**Lemma 4** (Bounded Hessian for smoothed densities). *Let $p_0$ be a probability density on $\mathbb{R}^d$ and let*

$$X_t = \mu_t X_0 + \sigma_t Z, \qquad Z \sim \mathcal{N}(0, I_d) \text{ independent of } X_0,$$

*with $\sigma_t \to 0$ and $\mu_t \to 1$ as $t \to 0$, and let $p_t$ be the density of $X_t$. Assume:*

*(A1) $p_0(x) > 0$ for all $x$ and $\ell_0(x) := \log p_0(x) \in C^2(\mathbb{R}^d)$;*

*(A2) $\nabla \ell_0$ is globally Lipschitz with constant $L < \infty$.*

*Then there exist $t_0 > 0$ and a constant $C < \infty$ such that*

$$\sup_{0 < t \le t_0} \mathbb{E}\big\|\nabla^2 \log p_t(X_t)\big\|_{\mathrm{op}} \;\le\; C,$$

*and we may take $C = (L + 1)/m^2$ for a suitable $m > 0$ depending only on $\mu_t$ near $t = 0$ (for instance, $m = e^{-t_0}$ for VP-SDE where $\mu_t = e^{-t}$, and $m = 1$ for VE-SDE where $\mu_t \equiv 1$).*

*Proof.* Write $\phi_\sigma$ for the Gaussian density with covariance $\sigma^2 I_d$. By (A2), the gradient $\nabla \ell_0$ is globally Lipschitz with constant $L$, so the Hessian exists everywhere and satisfies

$$\|\nabla^2 \ell_0(x)\|_{\mathrm{op}} \le L \qquad \text{for all } x \in \mathbb{R}^d.$$

Since $p_0(x) = \exp(\ell_0(x))$ and $\ell_0$ is continuous, $p_0$ is strictly positive and bounded on compact sets. From

$$\nabla p_0(x) = p_0(x) \nabla \ell_0(x), \qquad \nabla^2 p_0(x) = p_0(x)\big(\nabla^2 \ell_0(x) + \nabla \ell_0(x) \nabla \ell_0(x)^\top\big),$$

we also see that $p_0$, $\nabla p_0$, and $\nabla^2 p_0$ are bounded and uniformly continuous on compact subsets of $\mathbb{R}^d$.

Define

$$\tilde{\sigma}_t := \sigma_t / \mu_t, \qquad Y_t := X_0 + \tilde{\sigma}_t Z.$$

Then $Y_t$ has density $q_t = p_0 * \phi_{\tilde{\sigma}_t}$, the Gaussian smoothing of $p_0$ with bandwidth $\tilde{\sigma}_t \to 0$. Since Gaussian kernels with vanishing variance form an approximate identity, Folland (Folland, 1999, Theorem 8.14) gives

$$q_t \to p_0, \quad \nabla q_t \to \nabla p_0, \quad \nabla^2 q_t \to \nabla^2 p_0 \qquad \text{uniformly on compact sets.}$$

Because $p_0 > 0$, this uniform convergence implies that $q_t$ is bounded away from $0$ on compacts for all small $t$, and therefore the logarithms satisfy

$$\log q_t \to \log p_0, \qquad \nabla \log q_t \to \nabla \log p_0, \qquad \nabla^2 \log q_t \to \nabla^2 \log p_0 \quad \text{uniformly on compacts.}$$

In particular, since $\tilde{\sigma}_t \to 0$ and $q_t = p_0 * \phi_{\tilde{\sigma}_t}$, with $\phi_{\tilde{\sigma}_t}$ an approximate identity (Folland (Folland, 1999, Thm. 8.14)), we have

$$\nabla^2 \log q_t \longrightarrow \nabla^2 \log p_0 \quad \text{uniformly on compact subsets of } \mathbb{R}^d.$$

Hence, for any fixed radius $R > 0$, there exists $t_R > 0$ such that for all $t < t_R$ and all $\|y\| \leq R$,

$$\left\| \nabla^2 \log q_t(y) - \nabla^2 \log p_0(y) \right\|_{\text{op}} \leq 1.$$

By the triangle inequality, for all $\|y\| \leq R$ and all sufficiently small $t$,

$$\|\nabla^2 \log q_t(y)\|_{\text{op}} \leq \|\nabla^2 \log p_0(y)\|_{\text{op}} + \left\| \nabla^2 \log q_t(y) - \nabla^2 \log p_0(y) \right\|_{\text{op}}$$

$$\leq \sup_{\|z\| \leq R} \|\nabla^2 \log p_0(z)\|_{\text{op}} + 1.$$

Using the Lipschitz assumption (A2), we have

$$\sup_{\|y\| \leq R} \|\nabla^2 \log q_t(y)\|_{\text{op}} \leq L + 1, \qquad \text{for all sufficiently small } t.$$

We now relate $p_t$ and $q_t$. Since $X_t = \mu_t Y_t$, the change-of-variables formula gives

$$p_t(x) = \mu_t^{-d} q_t(x/\mu_t),$$

and hence

$$\nabla \log p_t(x) = \mu_t^{-1} \nabla \log q_t(x/\mu_t), \qquad \nabla^2 \log p_t(x) = \mu_t^{-2} \nabla^2 \log q_t(x/\mu_t).$$

Therefore,

$$\|\nabla^2 \log p_t(x)\|_{\text{op}} \leq \mu_t^{-2}(L+1) \qquad \text{for all small } t.$$

Since $\mu_t \to 1$, we may choose $t_0 > 0$ and $m > 0$ such that $m \leq \mu_t \leq 2$ for all $0 < t \leq t_0$. Hence,

$$\sup_x \|\nabla^2 \log p_t(x)\|_{\text{op}} \leq \frac{L+1}{m^2}, \qquad 0 < t \leq t_0,$$

and therefore

$$\mathbb{E}\|\nabla^2 \log p_t(X_t)\|_{\text{op}} \leq \frac{L+1}{m^2}.$$

Thus we may take $C = (L+1)/m^2$, and in particular

$$\sup_{0 < t \leq t_0} \mathbb{E}\|\nabla^2 \log p_t(X_t)\|_{\text{op}} < C,$$

so no blow-up occurs as $t \to 0$. $\qquad \square$

With Lemma 4, we can now sharpen the curvature term appearing in Theorem 3.4 for small times, replacing the worst-case $1/\tau$ behavior by a finite constant whenever $p_0$ has a Lipschitz score.

**Corollary B.1** (Refined local curvature control for Theorem 3.4). *Under the assumptions of Lemma 4, there exist $t_0 > 0$ and constants $m > 0$,*

$$C_0 = \frac{L+1}{m^2} < \infty,$$

*such that*

$$\sup_{0 < t \leq t_0} \|\nabla^2 \log p_t(x)\|_{\text{op}} \leq C_0 \qquad \text{for all } x \in \mathbb{R}^d.$$

*Consequently, in the Wasserstein bound of Theorem 3.4, the curvature matrices $H_j$ may be replaced by the sharper piecewise form*

$$H_j = \begin{cases} C_0 \, I_d, & 0 < \tau_j \leq t_0, \\[2mm] \dfrac{d}{\sigma_{\text{VP}}(\tau_j)^2} \, I_d, & \tau_j > t_0, \ \textit{VP-SDE}, \\[2mm] \dfrac{d}{\sigma_{\text{VE}}(\tau_j)^2} \, I_d, & \tau_j > t_0, \ \textit{VE-SDE}, \end{cases}$$

*where $\sigma_{\text{VP}}(\tau) = \sqrt{1 - e^{-2\tau}}$ and $\sigma_{\text{VE}}(\tau) = \sqrt{2\tau}$.*

## C   GAUSSIAN MIXTURE HESSIAN APPROXIMATION

**Hessian decomposition and responsibilities.**   For Gaussian mixtures

$$p_t(x) = \sum_{m=1}^{K} \pi_m \mathcal{N}(x; \mu_m(t), \Sigma_m(t)),$$

the (posterior) responsibility of component $m$ at location $x$ is

$$\gamma_m(x) := \frac{\pi_m \mathcal{N}(x; \mu_m(t), \Sigma_m(t))}{\sum_{\ell=1}^{K} \pi_\ell \mathcal{N}(x; \mu_\ell(t), \Sigma_\ell(t))}.$$

With this notation, the score Hessian admits the exact decomposition

$$\nabla^2 \log p_t(x) = -\sum_{m=1}^{K} \gamma_m(x) \, \Sigma_m(t)^{-1} + \mathrm{Cov}_{m\sim\gamma(\cdot|x)}[v_m(x)], \quad v_m(x) := \Sigma_m(t)^{-1}(\mu_m(t) - x).$$
$$(25)$$

**Separation regimes.**   Define the mean–separation surrogate

$$\delta_\mu(t) := \max_{m\neq n} \left\| \Sigma_m(t)^{-1/2}\big(\mu_m(t) - \mu_n(t)\big) \right\|,$$

and the covariance–separation surrogate

$$\delta_\Sigma(t) := \max_{m,n} \left\| \Sigma_m(t)^{1/2} \, \Sigma_n(t)^{-1} \, \Sigma_m(t)^{1/2} - I_d \right\|_{\mathrm{op}}.$$

We bundle them into a single small–separation parameter

$$\delta(t) := \max\big(\delta_\mu(t), \delta_\Sigma(t)\big).$$

We say *small separation* if $\delta(t) \ll 1$.

For large separation, define the logit margin

$$\kappa_t(x) := \min_{j\neq i^*(x)} \big(\ell_{i^*}(x) - \ell_j(x)\big), \qquad i^*(x) = \arg\max_i \ell_i(x),$$

where $\ell_i(x) = \log\pi_i - \frac{1}{2}\log\det(2\pi\Sigma_i(t)) - \frac{1}{2}(x - \mu_i(t))^\top \Sigma_i(t)^{-1}(x - \mu_i(t))$. We say *large separation* along a path $\{x_t\}$ if $\kappa_{\tau_k}(x_t) \geq \underline{\kappa} \gg 1$ for all $t \in [0,1]$.

The mixture Hessian can be approximated by surrogates of the form $-\sum_m w_m \Sigma_m^{-1}$ with different choices of weights $w_m$. A crude bound is always available by taking the prior weights $\pi_m$, but this ignores how the posterior responsibilities $\gamma_m(x)$ behave in different regimes. In the small–separation regime, the responsibilities remain close to the prior $\pi$, so the surrogate $\bar{H}_k = -\sum_m \pi_m \Sigma_m^{-1}$ achieves accuracy $O(\Lambda\delta(\tau_k))$. In the large–separation regime, the posterior mass concentrates sharply on one component, so a hard surrogate $\bar{H}_k = -\Sigma_{i^*}^{-1}$ is more appropriate, leading to exponential accuracy $O(\Lambda e^{-\underline{\kappa}})$. Accordingly, we analyze these three cases separately: a crude uniform bound (Lemma 5), a refined small–separation bound (Lemma 6), and a large–separation bound (Lemma 7).

**Lemma 5** (Crude uniform bound for surrogate Hessians). *Let*

$$H_k = \int_0^1 \nabla^2 \log p_{\tau_k}\big(y_k^{(0)} + t\Delta_k\big) \, dt, \qquad \bar{H}_k \in \Big\{ -\sum_{m=1}^{K} \pi_m \, \Sigma_m(\tau_k)^{-1}, \;\; -\Sigma_{i^*(y_k)}(\tau_k)^{-1} \Big\},$$

*where $i^\star(y_k) = \arg\max_{m\in[K]} \gamma_m(y_k; \tau_k)$, and set $\Lambda := \max_{m\in[K]} \|\Sigma_m(\tau_k)^{-1}\|_{\mathrm{op}}$. Then*

$$\mathbb{E}\big\| H_k - \bar{H}_k \big\|_{\mathrm{op}} \leq (d+2)\,\Lambda. \tag{26}$$

*Proof.* By the mixture Hessian identity,

$$\nabla^2 \log p_{\tau_k}(x) = -\sum_{m=1}^{K} \gamma_m(x; \tau_k) \, \Sigma_m(\tau_k)^{-1} + \mathrm{Cov}_{m\sim\gamma(\cdot|x;\tau_k)}\big[\Sigma_m(\tau_k)^{-1}(\mu_m(\tau_k) - x)\big].$$

Averaging along the segment $x_t := y_k^{(0)} + t\Delta_k$ and subtracting $\bar{H}_k$ gives

$$H_k - \bar{H}_k = \underbrace{-\int_0^1 \sum_{m=1}^K w_m(x_t)\,\Sigma_m(\tau_k)^{-1}\,dt}_{\text{term 1}} + \underbrace{\int_0^1 \mathrm{Cov}_{m\sim\gamma(\cdot|x_t;\tau_k)}[v_m(x_t)]\,dt}_{\text{term 2}},$$

where $v_m(x) = \Sigma_m(\tau_k)^{-1}(\mu_m(\tau_k) - x)$ and

$$w_m(x_t) = \begin{cases} \gamma_m(x_t;\tau_k) - \pi_m, & \text{if } \bar{H}_k = -\sum_j \pi_j \Sigma_j(\tau_k)^{-1}, \\ \gamma_m(x_t;\tau_k) - \mathbf{1}_{\{m=i^\star(y_k)\}}, & \text{if } \bar{H}_k = -\Sigma_{i^\star(y_k)}(\tau_k)^{-1}. \end{cases}$$

**Term 1** For any choice of $w_m$ above,

$$\Big\| \sum_{m=1}^K w_m(x_t)\,\Sigma_m(\tau_k)^{-1} \Big\|_{\mathrm{op}} \leq \sum_{m=1}^K |w_m(x_t)|\,\|\Sigma_m(\tau_k)^{-1}\|_{\mathrm{op}} \leq \|w(x_t)\|_1\,\Lambda.$$

In the mixture-weighted case, $\|w(x_t)\|_1 = \|\gamma(x_t;\tau_k) - \pi\|_1 \leq 2$.

In case where $\bar{H}_k = -\Sigma_{i^\star(y_k)}(\tau_k)^{-1}$, writing $m^\star = i^\star(y_k)$,

$$\|w(x_t)\|_1 = \sum_m |\gamma_m(x_t;\tau_k) - \mathbf{1}_{\{m=m^\star\}}| = 2\big(1 - \gamma_{m^\star}(x_t;\tau_k)\big) \leq 2.$$

Thus $\mathbb{E}\|\text{term 1}\| \leq 2\,\Lambda$.

**Term 2** Since covariance is PSD and $\|A\|_{\mathrm{op}} \leq \mathrm{tr}\,(A))$,

$$\big\|\mathrm{Cov}_{m\sim\gamma(\cdot|x)}[v_m(x)]\big\|_{\mathrm{op}} \leq \sum_{m=1}^K \gamma_m(x;\tau_k)\,\mathrm{tr}\,\big(\Sigma_m(\tau_k)^{-1}\big) \leq d\,\Lambda.$$

Integrating over $t \in [0,1]$ and taking expectation obtains $\mathbb{E}\|\text{term 2}\| \leq d\,\Lambda$.

Combining the bounds for the two terms gets $\mathbb{E}\|H_k - \bar{H}_k\|_{\mathrm{op}} \leq d\,\Lambda + 2\,\Lambda = (d+2)\Lambda$, which proves equation 26. $\qquad\square$

**Lemma 6** (Small separation bound). *In the setting of Lemma 5, assume the small–separation condition $\delta(\tau_k) = \max(\delta_\mu(\tau_k), \delta_\Sigma(\tau_k)) \ll 1$. Then*

$$\mathbb{E}\|H_k - \bar{H}_k\|_{\mathrm{op}} = O\big(\Lambda\,\delta(\tau_k)\big). \tag{27}$$

*Proof.* We have the decomposition

$$H_k - \bar{H}_k = \underbrace{-\int_0^1 \sum_m (\gamma_m(x_t) - \pi_m)\,\Sigma_m(\tau_k)^{-1}\,dt}_{\text{term 1}} + \underbrace{\int_0^1 \mathrm{Cov}_{m\sim\gamma(\cdot|x_t)}[v_m(x_t)]\,dt}_{\text{term 2}},$$

**Term 1.** Define the logits

$$\theta_m(x) := \log\pi_m - \tfrac{1}{2}\log\det(2\pi\Sigma_m) - \tfrac{1}{2}(x-\mu_m)^\top\Sigma_m^{-1}(x-\mu_m), \quad m = 1,\ldots,K,$$

where $\pi = (\pi_1,\ldots,\pi_K)$ are the mixture weights with $\pi_m > 0$ and $\sum_{m=1}^K \pi_m = 1$. Let $\gamma(x) = (\gamma_1(x),\ldots,\gamma_K(x))$ denote the posterior component weights ("responsibilities") at $x$. Then

$$\gamma(x) = \mathrm{softmax}(\theta(x)), \qquad \pi = \mathrm{softmax}(\theta^0), \quad \theta_m^0 := \log\pi_m.$$

The Jacobian of the softmax map is $J(\theta) = \mathrm{Diag}(\gamma) - \gamma\gamma^\top$, which satisfies $\|J(\theta)\|_{\mathrm{op}} \leq \tfrac{1}{2}$. By the mean value theorem,

$$\|\gamma(x) - \pi\|_2 \leq \tfrac{1}{2}\|\theta(x) - \theta^0\|_2.$$

Consequently,

$$\sum_{m=1}^{K} |\gamma_m(x) - \pi_m| = \|\gamma(x) - \pi\|_1 \leq \frac{\sqrt{K}}{2} \|\theta(x) - \theta^0\|_2.$$

Now, when $\delta(\tau_k) \ll 1$, the mixture parameters $(\mu_m, \Sigma_m)$ are close to some average $(\bar{\mu}, \bar{\Sigma})$. Writing $z = \bar{\Sigma}^{-1/2}(x - \bar{\mu})$, a Taylor expansion shows

$$|\theta_m(x) - \theta_m^0| \leq C\,\delta(\tau_k)\,(1 + \|z\|^2).$$

Hence

$$\sum_{m=1}^{K} |\gamma_m(x) - \pi_m| \leq C\,\delta(\tau_k)\,(1 + \|z\|^2).$$

Taking expectations gives the desired control:

$$\mathbb{E}\|\text{term 1}\| \leq O(\Lambda\,\delta(\tau_k)).$$

**Term 2** Fix $x$ and define, as above,

$$\mathrm{Cov}_{m \sim \gamma(\cdot|x)}[v_m(x)] = \mathbb{E}_{m \sim \gamma(\cdot|x)}\big[(v_m(x) - \bar{v}(x))(v_m(x) - \bar{v}(x))^\top\big], \qquad \bar{v}(x) = \mathbb{E}_{m \sim \gamma(\cdot|x)}v_m(x).$$

By PSD and $\|A\|_{\mathrm{op}} \leq \mathrm{tr}(A)$,

$$\big\|\mathrm{Cov}_{m \sim \gamma(\cdot|x)}[v_m(x)]\big\|_{\mathrm{op}} \leq \mathbb{E}_{m \sim \gamma(\cdot|x)}\|v_m(x) - \bar{v}(x)\|^2.$$

Again using the small–separation condition and the same $z = \bar{\Sigma}(\tau_k)^{-1/2}(x - \bar{\mu}(\tau_k))$, one has the component spread bound

$$v_m(x) - v_n(x) = \Sigma_m(\tau_k)^{-1}\big(\mu_m(\tau_k) - x\big) - \Sigma_n(\tau_k)^{-1}\big(\mu_n(\tau_k) - x\big)$$
$$= \big(\Sigma_m(\tau_k)^{-1} - \Sigma_n(\tau_k)^{-1}\big)\big(\mu_m(\tau_k) - x\big) + \Sigma_n(\tau_k)^{-1}\big(\mu_m(\tau_k) - \mu_n(\tau_k)\big),$$

then

$$\|v_m(x) - v_n(x)\| \leq \|\Sigma_m(\tau_k)^{-1} - \Sigma_n(\tau_k)^{-1}\|_{\mathrm{op}} \|\mu_m(\tau_k) - x\| + \|\Sigma_n(\tau_k)^{-1}\|_{\mathrm{op}} \|\mu_m(\tau_k) - \mu_n(\tau_k)\|,$$

which implies

$$\big\|\mathrm{Cov}_{m \sim \gamma(\cdot|x)}[v_m(x)]\big\|_{\mathrm{op}} \leq C^2\,\Lambda^2\,\delta(\tau_k)^2\,(1 + \|z\|^2).$$

Averaging over $x$ (hence $z$) and $t \in [0,1]$, and using $\mathbb{E}(1 + \|z\|^2) = O(1)$, we get

$$\mathbb{E}\|\text{term 2}\| \leq O(\Lambda\,\delta(\tau_k)^2).$$

Together with term 1, this yields equation 27. $\qquad \square$

**Lemma 7** (Large separation with hard surrogate). *Let*

$$H_k = \int_0^1 \nabla^2 \log p_{\tau_k}(x_t)\,dt, \qquad \bar{H}_k = \bar{H}_k^{\mathrm{hard}} = -\Sigma_{i^\star(y_k)}(\tau_k)^{-1},$$

*where* $x_t = y_k^{(0)} + t\Delta_k$ *and* $i^\star(x) = \arg\max_m \ell_m(x)$. *Assume a uniform logit margin* $\kappa_{\tau_k}(x_t) \geq \underline{\kappa} \gg 1$ *for all* $t \in [0,1]$. *Then*

$$\mathbb{E}\|H_k - \bar{H}_k^{\mathrm{hard}}\|_{\mathrm{op}} = O(\Lambda\,e^{-\underline{\kappa}}). \tag{28}$$

*Proof.* We have the decomposition

$$H_k - \bar{H}_k^{\mathrm{hard}} = -\underbrace{\int_0^1 \Big(\sum_m \gamma_m(x_t)\Sigma_m^{-1} - \Sigma_{i^\star(y_k)}^{-1}\Big)dt}_{\text{term 1}} + \underbrace{\int_0^1 \mathrm{Cov}_{m \sim \gamma(\cdot|x_t)}[v_m(x_t)]dt}_{\text{term 2}},$$

where $v_m(x) = \Sigma_m^{-1}(\mu_m - x)$ and $\Lambda := \max_m \|\Sigma_m^{-1}\|_{\mathrm{op}}$.

**Term 1**  Insert and subtract $\Sigma_{i^\star(x_t)}^{-1}$:

$$\sum_m \gamma_m(x_t)\Sigma_m^{-1} - \Sigma_{i^\star(y_k)}^{-1} = \sum_{m \neq i^\star(x_t)} \gamma_m(x_t)(\Sigma_m^{-1} - \Sigma_{i^\star(x_t)}^{-1}) + (\Sigma_{i^\star(x_t)}^{-1} - \Sigma_{i^\star(y_k)}^{-1}).$$

The first bracket is bounded by

$$\Big\| \sum_{m \neq i^\star(x_t)} \gamma_m(x_t)(\Sigma_m^{-1} - \Sigma_{i^\star(x_t)}^{-1}) \Big\|_{\mathrm{op}} \leq 2\Lambda \sum_{m \neq i^\star(x_t)} \gamma_m(x_t).$$

The uniform margin implies

$$\sum_{m \neq i^\star(x_t)} \gamma_m(x_t) \leq Ce^{-\kappa}. \tag{29}$$

The index mismatch contributes at most $2\Lambda Ce^{-\kappa}$. Hence

$$\mathbb{E}\|\text{term 1}\| \;\leq\; C_1\Lambda e^{-\kappa}.$$

**Term 2**  Expanding around $i^\star(x_t)$,

$$\|\mathrm{Cov}_{m\sim\gamma(\cdot|x_t)}[v_m(x_t)]\|_{\mathrm{op}} \leq \sum_{m \neq i^\star(x_t)} \gamma_m(x_t)\|v_m(x_t) - v_{i^\star(x_t)}(x_t)\|^2.$$

Since $\|v_m - v_{i^\star}\|^2 = O(\Lambda)$ under bounded moments, and apply equation 29, we obtain

$$\mathbb{E}\|\text{term 2}\| \;\leq\; C_2\Lambda e^{-\kappa}.$$

Adding both terms gives equation 28. □

## D  EXPERIMENTS

To further illustrate our theoretical results, we compare several step-size schedules using both the Wasserstein error bounds predicted by Theorem 3.2 and empirical performance obtained from sampler DDPM sampling (Ho et al., 2020). The data distribution is a symmetric one-dimensional Gaussian mixture,

$$p_0(x) = \tfrac{1}{2}\mathcal{N}(-1,1) + \tfrac{1}{2}\mathcal{N}(1,1).$$

In the experiments, the score function is implemented as

$$s(x,t) = \nabla \log p_t(x) + \|x\| + z,$$

where $z \sim \mathcal{N}(0,1)$ is Gaussian noise that simulates a synthetic score-estimation error. One may verify that this synthetic error satisfies both the Lipschitz continuity and the bounded second-moment conditions required in Assumption 1.

As shown in Figure 2, schedules that allocate *smaller* steps near the data end ($t \simeq 0$) achieve a *smaller* final $W_2$, consistent with our theoretical finding that error amplification is most sensitive in this region. Moreover, Theorem 3.2 produces bounds that preserve the same ordering across discretization schedules, accurately capturing the empirical behavior observed during sampling.

1188
1189
1190
1191
1192
1193
1194
1195
1196
1197
1198
1199
1200
1201
1202
1203
1204
1205
1206
1207
1208
1209
1210
1211
1212
1213
1214
1215
1216
1217
1218
1219
1220
1221
1222

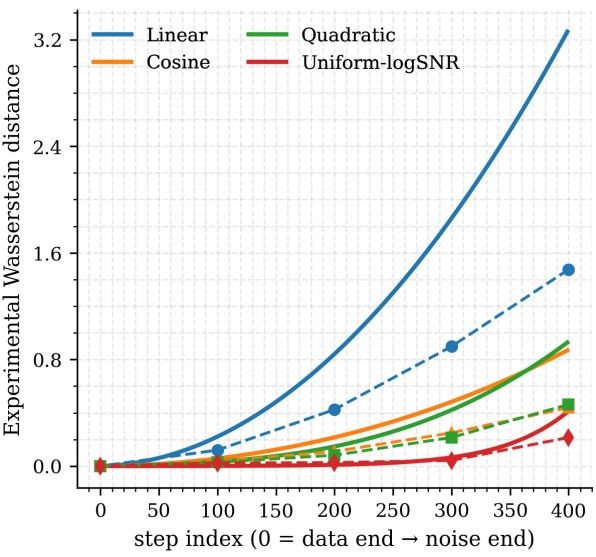

Figure 2: Theoretical $W_2$ error predicted by Theorem 3.2 (solid lines) together with empirical estimates (dashed lines). Schedules that place *smaller* steps near the data end ($t \simeq 0$) yield a *smaller* final $W_2$, confirming that error amplification is most sensitive in this region. The theoretical bounds closely track the empirical errors in both magnitude and trend, demonstrating the sharpness of our analysis.
