# OpenReview forum: "The Accumulation of Score Estimation Error in Diffusion Models"
_ICLR.cc/2026/Conference — Submitted to ICLR 2026_

### Official Review · Reviewer_sZFr · 2025-10-24

**Soundness:** 3
**Presentation:** 3
**Contribution:** 3
**Rating:** 8
**Confidence:** 3

**Summary:**

This paper analyzes the effect of score approximation error on the Wasserstein distance between a diffusion model's sample distribution and the ground truth target distribution, with a particular emphasis on its interaction with the time discretization of the reverse process. The authors first prove an exact Wasserstein bound for the simple case of a Gaussian target, which explicitly quantifies the error in the model distribution in terms of the discretization parameters, the structure of the target distribution, and the properties of the score estimation error. They then extend this exact analysis to the more representative case of a mixture-of-Gaussians target, and finally provide a looser bound on the Wasserstein error for general target distributions. The authors confirm that their bounds hold in practice by experimenting with a mixture-of-Gaussians target and show how their analysis can be used to explain the superior performance of SDE sampling.

**Strengths:**

- This is a well-written paper, and the authors did a good job of highlighting the important takeaways from their theorems and providing sufficient intuition.
- The key results are correct to my knowledge. However, I did not review the proofs in the appendices in exhaustive detail, and it is possible that I missed some errors.
- I appreciate that the authors prove results beyond the simple setting of a Gaussian target measure and consider multimodal targets such as mixtures of Gaussians. I also appreciate that the authors present Wasserstein bounds for the general case, even if these bounds are relatively loose.

**Weaknesses:**

My only quibble is that the error model (independent Gaussian perturbations) is somewhat unrepresentative of the score approximation errors that one would expect to observe in practice, which are more likely to arise from the inductive biases of the score model and the optimization algorithm. However, I understand that this analysis would be significantly more involved, and do not believe that this should be an obstacle to publication.

**Questions:**

- Are there plausible directions towards extending this paper's analysis to more general error models? I would be particularly interested in whether it would be possible to eventually relax the independence assumption on the score estimation error and analyze e.g. spatially-correlated errors.
- Can one obtain tighter bounds for the general case (as in Theorem 3.4) by approximating a general measure by a mixture of Gaussians and applying the Wasserstein bound for the latter case, or would the error from approximating a general measure by mixtures of Gaussians typically drown out the improved precision of the Wasserstein bound for mixtures of Gaussians?

---

> ### Author Response · Authors · 2025-11-22
> **Reply to the Weaknesses & Questions**
>
> **general error model**
>
> Thank you for the thoughtful comment. Our analysis does not rely solely
> on Assumption 1. Instead, we can work directly with the general score error
> $e(t,x)=s_\theta(x,t)-\nabla\log p_t(x)$ under two mild and standard
> conditions: (1) $e(t,\cdot)$ is Lipschitz in $x$ with constant $L_t$,and (2) its slice-wise second moment is bounded, a widely used
> $L^2$-type assumption on score-estimation error. We also verify these properties
> empirically in Gaussian-mixture experiments, confirming that the generalized
> assumption accurately reflects the behavior of a learned score model. Under this more realistic assumption, the reverse-time recursion maintains the
> same structure; the only change is that each linear factor
> $\alpha_i I_d+\beta_i H_i$ inside the amplification operator $G_i(H)$ is
> replaced by $\alpha_i I_d+\beta_i H_i+\beta_i L_{\tau_i} I_d$. Consequently,
> the overall bound $\mathbb{E}\|\Delta_K\|^2\le\sum_{i=0}^{K-1}\|G_i(H)\|^2 \varepsilon_{\tau_i}^2$ keeps the same form and dependence on curvature and
> step sizes. Thus, the results continue to hold under a general and more
> realistic model of score-estimation error, without requiring the
> Gaussian error model.
>
> **tighter bounds for the general case**
>
> One can in principle combine (i) the approximation error of representing a
> general distribution by a Gaussian mixture ($e_1$) with (ii) our derived bound for
> Gaussian mixtures ($e_2$) to obtain a bound for the general case. The  usefulness of
> this approach, however, depends on the regularity of the underlying
> distribution. When $p_0$ is smooth, the approximation error $e_1$ plus
> our mixture-based bound $e_2$ can indeed be tighter than the bound in Theorem 3.4.
>
> In contrast, for non-smooth distributions, even though $e_1$ can be made small, the Gaussian mixture components must use extremely small
> covariances to capture the sharp features of $p_0$, which makes $e_2$ large and may
> offset any advantage.
>
> It is worth noting that if $p_0$ satisfies a Lipschitz-score condition, then
> $\|\nabla^2 \log p_t(x)\|_{\mathrm{op}}$ remains bounded near $t=0$, eliminating
> the curvature blow-up in the general case and enabling substantially tighter bounds than Theorem 3.4. We will clarify this in the revised manuscript.

---

> > ### Comment · Reviewer_sZFr · 2025-11-26
> >
> > Thank you for your helpful response to my review. I particularly appreciate your clarifications regarding the possibility of obtaining tighter bounds by approximating general measures by Gaussian mixtures. I maintain my accept score for this paper.

---

### Official Review · Reviewer_5XYX · 2025-10-28

**Soundness:** 2
**Presentation:** 3
**Contribution:** 2
**Rating:** 4
**Confidence:** 3

**Summary:**

This paper studies how estimation errors in the score function affect the terminal distribution of the discretized reverse process in diffusion models. In particular, the authors assume that instead of numerically integrating the reverse SDE/probability flow (PF) ODE using the true score, we first additively perturb the true score with independent Gaussian noise $e_t \sim \mathcal{N}(\mu_t, \Sigma_t)$ at each timestep $t$. In all results, both the variance exploding (VE), and the variance preserving (VP) variants are considered. When the forward process is initialized from a Gaussian Mixture Model (the "data distribution" $p_0$), the authors provide a closed form expression for the Wasserstein distance between the terminal distributions of the numerical solutions of the VP/VE reverse SDE when using the true and perturbed scores.
However, since the pathwise Hessian average---a quantity required for the Wasserstein distance---is typically not accessible, the paper shows how the Wasserstein distance can still be bounded using two computationally tractable surrogates. This result is also extended to the PF-ODE.
When relaxing the distributional assumption on $p_0$, the authors are still able to provide a bound on the Wasserstein distance---albeit only with conservative guarantees, as the bound becomes vacuous as $t \to 0$.
The presented results are consistent with previous empirical observations regarding different discretization schedules, and the empirical advantage of SDE-based samplers over their PF-ODE counterparts.

**Strengths:**

+ **Motivation.** The work is well-motivated. Understanding the effect of discretization schemes in samplers that use inexact scores is crucial in developing more efficient and robust sampling algorithms for diffusion models, which is an important line of research in contemporary generative modeling.
+ **Presentation.** The main results are presented in a clear and unambiguous way. All assumptions are made explicit.
+ **Theoretical Soundness.** The paper seems theoretically sound and technically correct.
+ **Agreement with previous empirical findings.** The presented results are consistent with known empirical results, i.e., the empirical advantage of discretization schedules that are coarse at large $t$, and fine at $t \approx 0$, and that SDE-based samplers typically produce better samples than their PF-ODE-based counterparts (in the small-step regime).
+ **Closed Form Results.** Under the assumption that $p_0$ is Gaussian (or a GMM), the authors present exact, closed form results for the aforementioned Wasserstein distance. In practice, this distributional assumption is well justified: For example, training diffusion models on finite data (and truncating the reverse process such that the variance of the forward process is non-zero at $t=0$) gives rise to $p_0$ being a GMM (empirical, slightly smoothed data distribution).

**Weaknesses:**

+ **Practical applicability of Assumption 1**
	+ Since $s(x,t)$ is typically modeled with a neural network which is Lipschitz in $t$, the assumption of adding *independent* Gaussian perturbations $e_t$ seems strong.
	+ In practice, the score error will depend on the location $x$ (e.g., how far away $x$ is from the manifold), and will show strong temporal correlation.
+ **Assumption 1 vs. $L^2$-bounded score assumption**
	+ While Assumption 1 implies $L^2$-boundedness, the reverse is not true: The assumption of $L^2$-bounded scores does make distributional assumptions, and is more general than Assumption 1. It also allows for temporal correlation, error distributions that depend on $x$, and non-Gaussian error, making it more practically relevant than Assumption 1.
+ **Empirical Validation**
	+ The paper only provides a single experiment with a 1D GMM, where the score is perturbed when $t \in [0, 0.4T]$. However, even in this controlled toy experiment where Assumption 1 is true by construction, the deviation between the magnitudes of theoretical and empirical Wasserstein distance is non-negligible. Additional experiments with different index sets $I$ and different (high-dimensional) GMMs are missing. Such experiments would strengthen the practical validity of the theoretical results.
	+ Experiments for non-GMM $p_0$ are missing.
	+ The paper lacks experiments that investigate the applicability of Assumption 1 in more practical scenarios. For example, one could train a score-based model on a known GMM, and then investigate the error distribution.

+ **Notation & Typos**
	+ Notational inconsistency in Eq. 11: There's a squared $L_2$ norm around $\\beta_i \\exp(\\sum_{j=i+1}^{K-1} h^{\\leftarrow}\_j \\phi_{\\tau_j})$ in the first sum (although this is a scalar), while in the second sum, this term is just squared.
	+ Eq. 11: There's a superfluous closing bracket in the subscript of the first sum
	+ L178: "Eq. equation 8", L246: "Eq. equation 8", L284: "Eq. equation 12"
	+ Eq. 12 overloads $K$, where $K-1$ previously denoted the number of bins in $[0, T]$ (see L115)

**Questions:**

+ How reasonable is Assumption 1 in practice? Is there empirical evidence for it when considering trained diffusion models?
+ Can Assumption 1 be relaxed (temporal correlation, $x$-dependent error, etc.)?
+ How sensitive are the empirical results to the choice of $I$ and the dimensionality of $x$?
+ Are there connections between this work and Stochastic Gradient Langevin Dynamics (SGLD)?

---

> ### Author Response · Authors · 2025-11-22
> **Reply to the Weaknesses & Questions**
>
> **Practical applicability of Assumption 1:**
>
> Thanks for the insightful comment. The score error in practice can depend on
> $(x,t)$ and need not resemble independent Gaussian noise. Our analysis
> does not rely on this restrictive Gaussian score error Model. In the revision, we work
> directly with the general state-dependent error
> $e(t,x)=s_\theta(x,t)-\nabla\log p_t(x),$ under two mild conditions: (1) $e(t,\cdot)$ is Lipschitz in $x$ with constant $L_t$, and (2) its slice-wise second moment is bounded, $\mathbb{E}|e(t,x)|^2\le \varepsilon_t^2$, a widely used assumption for score-estimation error in $L^2$.
>
> Under this more realistic assumption, the reverse-time recursion maintains the
> same structure; the only change is that each linear factor
> $\alpha_i I_d+\beta_i H_i$ inside the amplification operator $G_i(H)$ is
> replaced by $\alpha_i I_d+\beta_i H_i+\beta_i L_{\tau_i} I_d$. Consequently,
> the overall bound $\mathbb{E}\|\Delta_K\|^2\le\sum_{i=0}^{K-1}\|G_i(H)\|^2 \varepsilon_{\tau_i}^2$ keeps the same form and dependence on curvature and
> step sizes. Thus, the results continue to hold under a general and more
> realistic model of score-estimation error, without requiring the
> Gaussian error model.
>
> **Assumption 1 vs. $L^2$-bounded score assumption:**
>
> As noted above, we do not rely on Assumption 1 in the revised analysis. Instead,
> we work directly with the general score error
> $e(t,x)=s_\theta(x,t)-\nabla\log p_t(x)$,
> under two mild and practically relevant conditions: (1) $e(t,\cdot)$ is
> Lipschitz in $x$ with constant $L_t$,  and (2) the slice-wise second moment is bounded,
> $\mathbb{E}\|e(t,x)\|^2\le \varepsilon_t^2$. This formulation is strictly more general than previous Assumption 1.
>
>  **Empirical Validation:**
>
> The theoretical result we derived is an upper bound. In deriving the surrogate
> curvature and operator-norm estimates, several relaxations were introduced to
> keep the analysis tractable, which naturally creates a gap between the theoretical
> and empirical Wasserstein distances. The score is perturbed only on
> $t\in[0,0.4T]$ because, in practice, the score error is known to be significantly
> larger near the data end, so our experiment focuses on this relative sensitive
> interval.
>
> In addition to the toy example, we have added a new experiment based on a learned
> score network rather than injected noise. We train a neural network to estimate
> the score of a (high-dimensional) Gaussian mixture using the standard DDPM
> objective and evaluate several sampling schedules via the Wasserstein distance. Across all tested schedules, we
> consistently observe that allocating smaller step sizes near $t\approx 0$ leads
> to smaller final $W_2$, matching the theoretical prediction.
>
> Moreover, prior work [1] on large-scale diffusion models (e.g., ImageNet 64×64 and
> CIFAR-10) has reported that cosine-type schedules outperform linear schedules
> under the same pretrained score model. This empirical results aligns with the
> preference suggested by our analysis and complements both the toy experiment and
> the learned-score experiment included in the revision.  **As for the  investigation of Assumption 1**,  in our experiments on Gaussian mixtures, we verify that both conditions  $e(t,\cdot)$ is
> Lipschitz in $x$ with constant $L_t$,  and (2) the slice-wise second moment is bounded hold for a learned score network. We will clarify this in the revised version.
>
> [1] Alexander Quinn Nichol and Prafulla Dhariwal. *Improved Denoising Diffusion Probabilistic Models*, 2021.
>
> **Notation & Typos:**
>
> Thanks for pointing these issues out — we have corrected the notation and typos in the revised version.
>
> **Q1-Q2:  Assumption 1**
>
> As noted above, our revised analysis does not only rely on Assumption 1. Instead, we
> work directly with the general score error $e(t,x)=s_\theta(x,t)-\nabla\log
> p_t(x)$ under two mild and practically relevant conditions: (1) $e(t,\cdot)$ is
> Lipschitz in $x$ with constant $L_t$, and (2) the slice-wise second moment is
> bounded. In our Gaussian–mixture experiments, we verify that both properties are
> satisfied by a learned score network, supporting the practical relevance of this
> assumption.
>
> **Q3:empirical results**
>
> Our experiments focus on choosing $I$ near the data end, since the score estimation error is empirically much larger in this region and therefore most relevant for studying error accumulation. We observe the same qualitative behavior in both the 1D toy setting and higher-dimensional Gaussian–mixture experiments with a learned score network. A more systematic investigation of how error accumulation scales with ambient dimension is an interesting direction for future work.
>
> **Q4: SGLD**
>
> SGLD analyzes how mini-batch gradient noise affects parameter-space posterior sampling, whereas our work analyzes how learned-score approximation error accumulates along the data-space reverse diffusion trajectory.

---

### Official Review · Reviewer_DRuC · 2025-10-30

**Soundness:** 3
**Presentation:** 2
**Contribution:** 2
**Rating:** 2
**Confidence:** 4

**Summary:**

This paper provides theoretical analysis on how score approximation errors in diffusion models (both VP/VE; SDE/ODE) propagates in the generation phase. Assuming time-dependent Gaussian score approximation errors, this paper quantifies the accumulation of score approximation error in Wasserstein-2 distance for data distributions that are Gaussian (Theorem 3.1), mixture of Gaussians (Theorems 3.2 and 3.3) and general distributions (Theorem 3.4).

The theoretical results can reveal known empirical findings in diffusion models, such as

(1) high-quality generation prefers smaller step-size towards the end of the generation phase;

(2) SDE sampler accumulates less score approximation errors than ODE sampler.

**Strengths:**

1. **Good motivation**: the effects of the score approximation errors in diffusion models are essential. To obtain theoretical understanding of how such errors accumulate help increase the generation quality, as well as understand memorization and generalization in diffusion models.

2. **Rigorous and complete analysis**: in the setting of the paper, the theoretical analysis is done rigorously from simple data distributions (Gaussian) to complicated ones (mixture of Gaussians), and to general ones.

**Weaknesses:**

1. **Issues on interpretation of the theoretical Results**:

    (1) *insufficient*: there are not many empirical evidences to explain the impact of the theoretical results, which makes people doubt how useful the proved theory is. The only numerical evidence in the main paper (Figure 1) is a toy example for mixture of Gaussians, which aims to illustrate that the empirical choices of small step-size at late-stage of generation can be explained by the theory. Even in this numerical example, it is not clear how the numerical errors in the right plot of Figure 1 are obtained.

    (2) *inaccurate*: one of the main claims in this paper is that the theory can explain the empirical success of cosine/uniform log-SNR schedules compared to linear ones, with numerical evidence provided. However, I find the claim is inaccurate. If the authors meant to say linear schedule is bad due to large score error accumulation, it is not very accurate since the discretization error analysis [1] also dislikes the linear schedule. If the authors meant that their theory can explain that log-SNR is optimal, as illustrated in Figure 1-Right, I feel that is also inaccurate: in practice, the log-SNR schedule is not optimal if we want to optimize training, which is empirically studied in [2] and theoretically studied in [3].

2. **Questionable setting**: the theoretical setting assumes the score approximation errors are Gaussians, uniform in space. Although this setting makes the analysis work, whether it is reasonable is questionable. My concern is that in this oversimplified setting, some interpretations of the proved results might be misleading if we use it to understand diffusion models in practice. What's mentioned in Weakness-1-(2) is a concrete example reflecting my concern.

[1] Chen, Hongrui, Holden Lee, and Jianfeng Lu. "Improved analysis of score-based generative modeling: User-friendly bounds under minimal smoothness assumptions." International Conference on Machine Learning. PMLR, 2023.

[2] Karras, Tero, et al. "Elucidating the design space of diffusion-based generative models." Advances in neural information processing systems 35 (2022): 26565-26577.

[3] Wang, Yuqing, Ye He, and Molei Tao. "Evaluating the design space of diffusion-based generative models." Advances in Neural Information Processing Systems 37 (2024): 19307-19352.

**Questions:**

1. The SDE (2) is written in the forward time. So $Y_t\sim p_t$ exactly. If we need $Y_t\sim p_{T-t}$, we need to write the SDE in reverse time.

2. From the theoretical results, such as (11), what's the exact optimal time schedule to minimize the $W_2$ score error accumulation? For the mixture of Gaussians case, does the optimal schedule depend on the two separation regimes?

3. How are the empirical errors in Figure 1-Right obtained? Is the empirical generation training-free?

4. In the section **Future Work**, the authors mentioned end-to-end error analysis of diffusion models. There are existing work doing this, and especially focusing on the optimal (time/variance/weight) schedules. See [3].

[3] Wang, Yuqing, Ye He, and Molei Tao. "Evaluating the design space of diffusion-based generative models." Advances in Neural Information Processing Systems 37 (2024): 19307-19352.

---

> ### Author Response · Authors · 2025-11-22
> **Reply to the Weaknesses**
>
> **Issues on interpretation of the theoretical Results:**
>
> **(1) insufficient:**
>
> The numerical errors in the right plot of Figure 1 are obtained by injecting controlled artificial noise into the *ground-truth* score function, so that we can directly isolate and visualize how error at each time slice propagates through the reverse dynamics.
>
> Beyond this illustrative setup, we have added a new set of experiments using a *learned score network** rather than injected noise. We train a neural network to estimate the score of a Gaussian mixture using the standard DDPM objective, and for this learned model we compare several sampling schedules by measuring the Wasserstein distance. Across all tested schedules, we observe a consistent pattern: schedules that use smaller step sizes near the data end ($t\approx 0$) yield smaller final $W_2$ distance, matching the theoretical prediction that early-time steps are the most sensitive. Moreover, prior work[1] on large-scale diffusion models (e.g., ImageNet 64×64 and CIFAR-10) indicates that cosine schedule provides better sampling performance than linear schedule, which is consistent with the preference suggested by our analysis. This provides empirical support based on a *real* learned score function, complementing the toy example in the main paper.
>
> [1] Alexander Quinn Nichol and Prafulla Dhariwal. *Improved Denoising Diffusion Probabilistic Models*, 2021.
>
> **(2) Inaccurate**
>
> We appreciate the reviewer’s detailed comments. Our theoretical results focus on a specific and orthogonal question: **given a fixed, trained score network**, how does the *score error* accumulate under different time-discretization
> schedules and affect the final distributional error measured in $W_2$?  Our analysis is therefore not concerned with the *discretization error* of the numerical solver, nor with identifying which schedule is optimal for **training**.
>
> In this sense, our conclusions are complementary to prior work. The discretization analysis in [1] also explains why the linear schedule is suboptimal, but it does so from the viewpoint of the discretization error, whereas our bound highlights the amplification
> of score-estimation error. So we address different sources of error.
>
> Regarding [2] and [3], these works study which noise schedule yields the best **training** behavior. This is a distinct problem from ours: we do not make claims about the optimal training schedule, nor do we argue that a log-SNR schedule is optimal for training. Our theory only characterizes how pre-existing score errors propagate under a given discretization schedule.
>
> **Questionable setting:**
>
> In fact, our
> analysis does not rely on Gaussian score error parameterization. We have updated the theory to work directly with the general state-dependent error
> $
> e(t,x)=s_\theta(x,t)-\nabla\log p_t(x),
> $
> under two mild conditions: (1) $e(t,\cdot)$ is Lipschitz in $x$ with constant $L_t$; and (2) its slice-wise second moment is bounded, $\mathbb{E}|e(t,x)|^2\le \varepsilon_t^2$, a widely used assumption for score-estimation error in $L^2$.
>
> Under this  assumption, the reverse-time recursion keeps the
> same form; the only change is that each linear factor
> $\alpha_i I_d + \beta_i H_i
> $
> inside the amplification operator $G_i(H)$ is replaced by
> $
> \alpha_i I_d + \beta_i H_i + \beta_i L_{\tau_i} I_d.
> $
> Consequently, the overall propagation bound remains
> $
> \mathbb{E}\|\Delta_K\|^2 \le
> \sum_{i=0}^{K-1}\|G_i(H)\|^2\varepsilon_{\tau_i}^2,
> $
> with the same structure and the same dependence on curvature and step sizes. Thus our results continue to hold under a general and more realistic score error assumption model.

---

> ### Author Response · Authors · 2025-11-22
> **Reply to Questions**
>
> **Q1: SDE**
>
> In the paper, Equation (2) refers to the time-reversed SDE evolving from $T$ to $0$, so that the process satisfies $Y_t \sim p_{T-t}$ along its trajectory. To avoid any potential ambiguity, we will include an explicit statement clarifying the time direction in the updated manuscript.
>
> **Q2: optimal time schedule**
>
> In particular, our bound shows that regions with large score error $\varepsilon_t$ and strong amplification $\exp\big(\int_0^t \phi(s)\,ds\big)$, as in (11), ccontribute most to error growth. Hence, an optimal schedule would use smaller steps where these terms are large and larger steps where they are small — typically, small steps near the data end and large steps near the noise end. The optimal allocation of step sizes therefore depends on problem-specific quantities such as the score-error $\varepsilon_t$ and the curvature of the forward marginals $H_t$.
>
> For the mixture-of-Gaussians case, the separation regime affects the tightness of our theoretical bound, since the surrogate curvature (mixture-averaged vs.\ dominant-component) is determined by whether the distribution lies in the small- or large-separation regime. As a result, different separation regimes influence how the curvature behaves and thus modify the preferred step-size allocation.
>
> **Q3: empirical errors**
>
> As noted above, the empirical errors in Figure 1-Right are obtained by injecting artificial noise into the ground-truth score function, which allows us to isolate and visualize how slice-wise score errors propagate through the reverse dynamics without confounding factors. In the revised version, we also include results based on a trained score network to complement this illustrative, training-free experiment.
>
> **Q4:  Future Work**
>
> Regarding [3], this work provides an important step toward end-to-end analysis by
> establishing convergence of the *global* denoising score matching objective
> $\bar L_{\mathrm{em}}(\theta),$
> which aggregates the training loss across all time steps. Our setting is
> different: we analyze how the score error at each individual time slice
> accumulates during sampling under a chosen discretization schedule. To interface
> directly with our accumulation bound, one would need time-resolved training-error
> estimates.

---

> > ### Comment · Reviewer_DRuC · 2025-11-23
> >
> > I really appreciate the authors' clarifications on my questions and concerns.
> >
> > It is nice to see experimental results with learned scores and extended theory with score errors beyond Gaussian. However, I couldn't find the new experiments using the learned scores. It would be great if the authors could update the manuscript with the added part highlighted.
> >
> > Regarding the new Theorem 3.4, I would like to understand the sharpness of the bound. It looks like the dimension-dependency is $d^{2K}$ with $K$ the number of generation steps. This is quite bad. As a comparison, [1] considers similar score error assumptions, even though with the KL metric rather than $W_2$, their score-error accumulation bound is of order $T\varepsilon^2$ which won't blow-up so badly as $d$ increases. If we check the simplified version of Theorem 3.4 for Gaussians (Theorem 3.2) and MoGs (Theorem 3.2), there is no such dimension-dependency issue either. Therefore, I am wondering if the bound in Theorem 3.4 is too loose and if it can be improved by using the techniques in [1].
> >
> > [1]: Benton, Joe, et al. "Nearly $ d $-linear convergence bounds for diffusion models via stochastic localization."

---

> > > ### Author Response · Authors · 2025-11-25
> > > **Relpy to dimension-dependency**
> > >
> > > We thank the reviewer for pointing this out. In the revised manuscript, we have replaced **Figure 1 on page 7** with new experiments based on learned score networks (trained via DDPM). We have also **updated the theoretical sections** accordingly to reflect the new assumptions.
> > >
> > > Regarding the dimension-dependency, the analysis of [1] follows a different mechanism. Using a Girsanov-type representation, they express the continuous–discrete KL discrepancy as
> > > $$
> > > \mathrm{KL}(p_T \,\|\, p_T^{\mathrm{disc}}) \le
> > > \sum_{k=0}^{K-1} \int_{t_{k-1}}^{t_k} E \|\nabla\log p_{t_k}(X_{t_k}) - s_\theta(X_{t_k},t_k)\|^2 dt.
> > > $$
> > > Their analysis first applies the triangle inequality to decompose the
> > > integrand in the KL representation. This yields two distinct contributions:  (i) a discretization error term $
> > > \sum_k E\|\nabla\log p_{t_k}(X_{t_k})-\nabla\log p_t(X_t)\|,$ and (ii) a score-estimation error term $\sum_k E \|\nabla\log p_t(X_t)-s_\theta(X_{t_k},t_k)\|.$  Under the usual $L^2$ score-error assumption, the second term contributes $O(T\,\varepsilon^2)$. In other words, [1] does not examine the mechanism by which score errors accumulate throughout the reverse process. Their score-error term is obtained from an assumed $L^2$ score estimation error.
> > >
> > > Our work, in contrast, explicitly analyzes how score errors propagate through the discretized VP/VE reverse dynamics:
> > > $\Delta_{k+1}=M_k\Delta_k+\beta_k e(\tau_k,y_k), \Delta_K=\sum_i\Bigl(\prod_{j>i}M_j\Bigr)\beta_i e(\tau_i,y_i^{(0)})$.
> > > This introduces multiplicative amplification through $M_{op}$. In the distribution-free setting, the curvature of $p_t$ satisfies
> > > $
> > > E\|\nabla^2\log p_t(X_t)\|_{op} \le d/\sigma(t)^2,
> > > $ so a dependence on the dimension naturally appears in our bound under such general assumptions. In structured settings such as Gaussians or Gaussian mixtures, the Hessian is uniformly bounded and the dimension factor disappears (Theorems 3.2–3.3).
> > >
> > > In summary, our analysis targets a different question from [1] by explicitly characterizing how score errors propagate through the reverse dynamics. Developing techniques that reduce this dimension dependence under fully distribution-free assumptions is an interesting direction for future work.

---

### Official Review · Reviewer_sdxG · 2025-11-01

**Soundness:** 2
**Presentation:** 2
**Contribution:** 2
**Rating:** 2
**Confidence:** 4

**Summary:**

The authors analyze the accumulation of score estimation error in diffusion models. The paper derives non-asymptotic Wasserstein bounds for the error between the generated distribution and a baseline distribution, assuming the score error is an independent Gaussian perturbation at each step. The analysis is first performed for data drawn from a Gaussian and Gaussian mixture (GM) distribution, yielding bounds that depend on the discretization schedule and the properties of the score's Hessian. A more general, but admittedly looser, bound is provided for arbitrary data distributions. The authors use this framework to theoretically justify the empirical success of schedules that use smaller steps near the data end ($t \approx 0$) and to argue that SDE samplers accumulate less error than ODE samplers.

**Strengths:**

* The paper addresses the impact of **score estimation error**, which is a significant factor in model performance but less analyzed than discretization error.
* The analysis provides a tractable, closed-form Wasserstein error bound for the simplified case of a Gaussian initial distribution (Theorem 3.1).
* The theoretical results, particularly for the Gaussian and GM cases, provide a formal explanation for the empirically-observed sensitivity to step-size allocation, corroborating the strategy of refining steps near the data end.

**Weaknesses:**

### Weaknesses

*  **Assumption 1 is fundamentally flawed and unrealistic.** The paper models the score estimation error $e_t$ as an *independent Gaussian perturbation* $\mathcal{N}(\mu_t, D_t D_t^\top)$ injected at each reverse step. Formally, the sentence "independent at each time instant" is not meaningful, as the authors would need to consider a white noise process with a much more complex stochastic calculus machinery. Even neglecting this, the true error $s_\theta(x,t) - \nabla \log p_t(x)$ is  an highly structured function of the state $x$, time $t$, and network parameters $\theta$. Consequently, modeling it as independent additive noise invalidates any claim of practical relevance. The analysis is for a system under *external* Gaussian noise, not for a system using an *imperfect* learned function.

* **Notation is inconsistent** is \mu the mean of the error vector, or is it the mean for the other considered noise schedules? Also (referring to line 606, proof of theorem), what does it mean "e... is the (zero–mean) score approximation error with mean ..."?
* **The general-distribution bound is useless.** The authors' own analysis of Theorem 3.4 admits it is "conservative" and "looser". Crucially, they state that the Hessian term $H$ "explode[s]" as $\tau \to 0$ (the data end). Given that the paper's primary conclusion is the critical importance of controlling error in this exact region , a bound that explodes there is not useful.
* **The tractable bounds (Thm 3.1, 3.2) apply only to toy distributions.** The main "sharp" results rely on the data being a single Gaussian or a Gaussian Mixture. This is analytically convenient but completely unrepresentative of the high-dimensional, complex-manifold data (like images) on which diffusion models are actually used.
* **Empirical validation is weak and self-referential.** The experiment in Figure 1  is conducted on a 1D Gaussian mixture and, most importantly, injects an artificial Gaussian error $e_t \sim \mathcal{N}(1,1)$. This experiment does not validate the theory on a real problem; it merely simulates the unrealistic conditions of Assumption 1. There is no evidence shown that these bounds or schedule rankings hold when using the *actual* error from a trained score network on a high-dimensional task.

**Questions:**

1.  **Regarding Assumption 1:** How can you justify modeling the score error?T he true error is the function $s_\theta(x_k, \tau_k) - \nabla \log p_{\tau_k}(x_k)$, which is highly correlated with $x_k$. Does this assumption not fundamentally misrepresent the problem?
2.  **Regarding Theorem 3.4:** You state that the general-distribution bound relies on a Hessian term that "explode[s]" near the data end ($\tau \approx 0$) and that the bound is "conservative". Since your main practical insight is the need for fine-grained steps in this exact region, what is the practical value of this general bound if it fails to be informative in the most critical regime?
3.  **Regarding Figure 1:** Your empirical test uses an *injected* Gaussian error $e_t \sim \mathcal{N}(1,1)$ on a 1D GMM, rather than the true error from a trained network. Can you provide any evidence that your theoretical bounds or the resulting schedule rankings (Linear vs. Cosine, etc.) hold in a realistic, high-dimensional setting using the *actual* error $s_\theta(x,t) - \nabla \log p_t(x)$?
4.  **Regarding SDE vs. ODE:** Your argument for SDE superiority rests on a comparison of linearized amplification factors. How does this simplistic comparison justify the strong claim that SDE samplers "accumulate less error overall"?

---

> ### Author Response · Authors · 2025-11-22
> **Reply to the Weaknesses 1-4**
>
> **Assumption 1 is fundamentally flawed and unrealistic.**
>
> Thanks for the insightful comment. We agree that the score error may depend on $(x,t)$. In fact, our
> analysis does not rely on this restrictive Gaussian error assumption. We have updated the theory to work directly with a general state-dependent error
> $
> e(t,x)=s_\theta(x,t)-\nabla\log p_t(x),
> $
> under two mild conditions: (1) $e(t,\cdot)$ is Lipschitz in $x$ with constant $L_t$; and (2) its slice-wise second moment is bounded, $\mathbb{E}|e(t,x)|^2\le \varepsilon_t^2$, a widely used assumption for score-estimation error in $L^2$.
>
> Under this assumption, the reverse-time recursion keeps the
> same form; the only change is that each linear factor $\alpha_i I_d + \beta_i H_i$
> inside the amplification operator $G_i(H)$ is replaced by $\alpha_i I_d + \beta_i H_i + \beta_i L_{\tau_i} I_d$. Consequently, the overall propagation bound remains $\mathbb{E}\|\Delta_K\|^2 \le \sum_{i=0}^{K-1}\|G_i(H)\|^2 \varepsilon_{\tau_i}^2,$
> with the same structure and the same dependence on curvature and step sizes. Thus our results continue to hold under a general and more realistic score error assumption model.
>
> **Notation is inconsistent.**
>
> To clarify, in our notation $\mu_\tau$ represents the mean of the score approximation error at time $\tau$ in our previous Gaussian Score error model. The phrase “(zero-mean)” in line 606 was misleading in this context, and we have corrected it with new score approximation model in the revised manuscript to ensure consistency.
>
> **The general-distribution bound is useless.**
>
> Theorem 3.4 is not intended to yield a tight numerical estimate near the data end. Its role is to characterize the **distribution-free baseline**, where no structural assumptions are made on $p_0$. The blow-up in Theorem 3.4 follows solely from the worst-case Hessian bound for an arbitrary distribution (Lemma 1). When no regularity assumptions are imposed on $p_0$, the forward diffusion provides only limited smoothing as $t\to0$, so in the distribution-free setting, it is possible to encounter the case where the curvature of $p_t$ reaches $\Theta(1/t)$ as $t\to0$ (Lemma 1). Consequently, the data end naturally becomes the most sensitive region.
>
> When even mild smoothness is present, the result changes substantially. As shown in [1], if $p_0$ has an $L$–Lipschitz score, then the forward marginals $p_t$ inherit uniformly Lipschitz scores for sufficiently small $t$. Equivalently, $\|\nabla^2\log p_t(x)\|_{op}$
> remains bounded near $t=0$, eliminating the $1/t$–type curvature growth observed in the worst case. This yields significantly milder bounds, fully consistent with our Gaussian and Gaussian-mixture analyses, where the curvature can be controlled explicitly.
>
> For these reasons, Theorem 3.4 is stated deliberately under no structural assumptions: it provides a worst-case baseline that clarifies what can be concluded for the distribution-free case:  the data-end region is inherently more sensitive to score-estimation errors.
>
> [1] Chen et al. (2023). *Improved analysis of score-based generative modeling: User-friendly bounds under minimal smoothness assumptions.* International Conference on Machine Learning. PMLR, 2023.
>
> **The tractable bounds (Thm 3.1, 3.2) apply only to toy distributions.**
>
> Our Gaussian and Gaussian-mixture results are not meant to approximate real image distributions, but to provide structured settings where the curvature of the forward marginals, $\nabla^2 \log p_t,$ can be well bounded and explicitly controlled. This allows us to examine early-time error amplification clearly.  These settings serve as clean benchmarks that help reveal the underlying mechanism.
>
> Importantly, our framework is not restricted to Gaussian or mixture distributions. As noted above, if the data distribution $p_0$ has an $L$-Lipschitz score, then the forward marginals $p_t$ also maintain  bounded-curvature for sufficiently small $t$. In this regime, our analysis applies in the same way: the curvature remains controlled near the data end, and the error propagation behaves similarly to the Gaussian and Gaussian-mixture cases.

---

> ### Author Response · Authors · 2025-11-22
> **Reply to the Weakness 5 & Questions**
>
> **Empirical validation is weak and self-referential.**
>
> To address this concern, we added a new set of experiments based on a **learned score network** rather than injected artificial noise. Specifically, we train a neural network to estimate the score of a Gaussian mixture using the standard DDPM objective. For this learned model, we compare multiple sampling schedules by measuring the $W_2$ distance between the generated samples and the true target distribution. Across all tested schedules, we consistently observe that schedules that allocate smaller step sizes near the data end $(t \approx 0)$ produce a  smaller final $W_2$ distance, which aligns exactly with our theoretical prediction about early-time sensitivity. In particular, cosine schedules have been
> observed to outperform linear schedules in large-scale DDPMs. For example, [1] evaluate DDPM with different schedules and report the FID:
> | Dataset | Linear Schedule | Cosine Schedule |
> |--------|-----------------|-----------------|
> | ImageNet 64×64 | **31.3** | **27.0** |
> | CIFAR-10 | **3.05** | **2.90** |
>
> This empirical evidence aligns with the preference predicted by our  analysis.
>
> [1] Alexander Quinn Nichol and Prafulla Dhariwal. *Improved Denoising Diffusion Probabilistic Models*, 2021.
>
> **Q1: Regarding Assumption 1**
>
> As we answered above, we can generalize the score error assumption to $
> e(t,x)=s_\theta(x,t)-\nabla\log p_t(x),
> $
> under two mild conditions: (1) $e(t,\cdot)$ is Lipschitz in $x$ with $L_t$, and  (2) its second moment is bounded,
> $\mathbb{E}\|e(t,x)\|^2\le \varepsilon_t^2$. Our results continue to hold under this general and more realistic score error assumption model.
>
> **Q2: Regarding Theorem 3.4**
>
> As discussed in our response above, the behavior of Theorem 3.4 near the data end is a direct consequence of making no assumptions at all on the data distribution $p_0$. In this fully distribution-free setting, the curvature of the forward marginals  $\log p_t$ can  reach the order $1/t$ as $t \rightarrow 0$, as shown in Lemma 1. The theorem therefore reflects the true worst-case geometry. In other words, Theorem 3.4 is valuable as a worst-case baseline explaining why the data end requires careful treatment, while our structured analyses and smoothness-based conditions show how the same conclusions become quantitatively sharp when $p_0$ satisfies mild regularity.
>
> **Q3:Regarding Figure 1**
>
> We have evaluated the actual score error $s_\theta(x,t)-\nabla\log p_t(x)$ in a Gaussian–mixture setting, where the measured score error accumulation agrees qualitatively with the trend predicted by our theoretical bound. Moreover, prior work on large-scale diffusion models (e.g., ImageNet 64×64 and CIFAR-10) indicates that cosine schedule provides better sampling performance than linear schedule, which is consistent with the trend suggested by our analysis.
>
> **Q4: Regarding SDE vs. ODE**
>
> Our theoretical analysis is derived in the Gaussian setting, where the Hessian admits an explicit form that allows us to compare the linearized amplification factors. Since $amp_{SDE}(t) < amp_{ODE}(t)$$, the SDE sampler accumulates less error for small steps: the amplification factors multiply across steps, so smaller per-step amplification directly translates into smaller total error. For general distributions, the Hessian may vary in sign and magnitude, and no universal ordering between amplification factors can be established. To avoid ambiguity, we will make the scope of our SDE–ODE comparison more explicit in the revised manuscript.

---

### Meta-Review · Area_Chair_Y8aJ · 2025-12-24

**Summary:**

Reviewer concerns included the following points.
- Initially, the paper assumed that the score errors are Gaussian. This overly stringent assumption has been removed and replaced with more standard score error assumptions, so this concern is addressed.
- The experimental results were lacking. In response, the authors added experiments with real trained score networks. From my own reading, this was not especially convincing. The main prediction offered by the theory here seems to be that one should take smaller step sizes at later times in the reverse diffusion, but this conclusion has already been arrived at by an abundance of works, both empirical and theoretical. The proposed experiments do not really demonstrate whether the derived bounds in this work have any value.
- Reviewer DRuC raised the issue that the stated bounds seem to be quantitatively quite poor, typically scaling exponentially in various parameters like the dimension. I personally share this concern; although the final theorem looks neat, all of the important dependencies are hiding in the definition of $G_i(H)$. These bounds are not really evaluated on any concrete examples, nor are they compared to prior works in the literature. In my understanding, these bounds are simply worse than the existing KL-based ones.

I would also like to raise the point that it is dangerous to make inferences based on comparisons of potentially loose upper bounds; for example, it is not possible to claim that the SDE attenuates errors less, simply because one upper bound is smaller than another upper bound, neither of which may resemble the truth.

The reviewers initially gave rather poor scores, with the exception of Reviewer sZFr who gave a score of 8. After the rebuttal phase, it is possible that the scores would increase slightly due to having addressed the concern about the Gaussian score error assumption, but it is unlikely that the paper would have met the bar for acceptance. Based on this and my own informed opinion, I mark this paper for rejection.

**Reviewer Concerns:**

The concern regarding the Gaussian score error assumption was resolved. The others are outstanding.

**Reviewer Scores:**

Reviewers sdxG, 5XYX would potentially have raised their scores slightly, as the Gaussian score error assumption has been relaxed. But likely not enough to advocate for acceptance.

Reviewer DRuC engaged in discussion and did not seem convinced; likely, the reviewer's score would have remained the same.

Reviewer sZFr was satisfied with the paper/rebuttal and kept the score the same.

---

### Decision · Program_Chairs · 2026-01-26

Reject